# Snowmelt-mediated isotopic homogenization of shallow till soil

**Filip Muhic**[1]**, Pertti Ala-Aho**[1]**, Matthias Sprenger**[2]**, Björn Klöve**[1]**, and Hannu Marttila**[1]

[1]Water, Energy and Environmental Engineering Research Unit, University of Oulu, Oulu, Finland
[2]Earth and Environmental Sciences, Lawrence Berkeley National Laboratory, Berkeley, CA, USA

**Correspondence:** Filip Muhic (filip.muhic@oulu.fi)

**Abstract.** The hydrological cycle of sub-arctic areas is dominated by the snowmelt event. An understanding of the mechanisms that control water fluxes during high-volume infiltration events in sub-arctic till soils is needed to assess how future changes in the timing and magnitude of snowmelt can affect soil water storage dynamics. We conducted a tracer experiment in which deuterated water was used to irrigate a plot on a forested hilltop in Lapland, tracked water fluxes of different mobility and monitored how the later snowmelt modifies the labelled soil water storage. We used lysimeters and destructive soil coring for soil water sampling and monitored and sampled the groundwater. Large spatiotemporal variability between the waters of different mobility was observed in the subsurface, while surface water flow during the tracer experiment was largely controlled by a fill-and-spill mechanism. Extensive soil saturation induced the flow of labelled water into the roots of nearby trees. We found that labelled water remained in deeper soil layers over the winter, but the snowmelt event gradually displaced all deuterated water and fully homogenized all water fluxes at the soil–vegetation interface. The conditions required for the full displacement of the old soil water occur only during a snowmelt with a persistently high groundwater table. We propose a conceptual model where infiltration into the soil and eventual soil water replenishment occur in three stages. First, unsaturated macropore flow is initiated via the surface microtopography and is directed towards the groundwater storage. The second stage is characterized by groundwater rise through the macropore network, subsequent pore water saturation and increased horizontal connectivity of macropores. Shallow subsurface lateral fluxes develop in more permeable shallow soil layers. In the third stage, which materializes during a long period with a high groundwater table and high hydrological connectivity within the soil, the soil water is replenished via enhanced matrix flow and pore water exchange with the macropore network.

## 1 Introduction

Soil water storage plays a vital role in sustaining the water cycle and regulating eco-hydrological processes at the soil–vegetation–atmosphere continuum (Sprenger et al., 2016). It acts as a link between precipitation and groundwater recharge, while soil moisture stored in the vadose zone provides the water for plant transpiration (Brooks et al., 2015). Soil moisture storage within the vadose zone is controlled by highly spatially and temporally variable mixing and transport processes, which are difficult to disentangle (Kampf et al., 2015; Penna et al., 2009; Vereecken et al., 2015). Therefore, the observed soil water patterns are often difficult to reproduce through modelling approaches, and a better conceptualization of soil water mixing is needed (Kuppel et al., 2018).

Studies that utilize stable isotopes of water to track water storages and fluxes at the soil–vegetation interface are becoming more frequent, as stable isotopes of water have provided new insights into soil water mixing and transport processes and the interaction between water fluxes that occupy or move through soil pores of different sizes (Good et al., 2015; McDonnell, 2014). The mixing dynamics of soil matrix water, usually moving slowly via matrix flow, and macropore water, infiltrating at a much higher velocity to deeper soil layers, influence infiltration patterns, soil moisture redistribution in unsaturated soil and the amount of water available to plants during the growing season (Phillips, 2010). Plant water uptake, one of the most important factors

in ecosystem functioning, can be dependent on both soil water compartmentalization (Brooks et al., 2010) and seasonal availability (Allen et al., 2019). On the other hand, soil water dynamics are also influenced by the vegetation, as soil water content and isotopic variability can be dependent on the plant cover (Oerter and Bowen, 2019), meaning that both soil and stem water observations are required to understand the fate of the water that infiltrates the soil.

Current studies on soil water mixing based on isotope data are geographically biased towards temperate climates (Tetzlaff et al., 2015), and more high-quality data sets from other climate regions are needed. A better understanding of subsurface water pathways is especially important in snow-dominated northern areas, where both current and predicted warming rates are highest (Hassol, 2004; Post et al., 2009). The higher latitudes are particularly sensitive to climate change through increased land surface temperatures (Hartmann et al., 2013), altered snow cover durations and amounts (Groisman et al., 1994; Pulliainen et al., 2020; Stiegler et al., 2016), and modifications of the freeze–thaw cycles (Hatami and Nazemi, 2022; Henry, 2008), rainfall and snowfall distributions (Bintanja and Andry, 2017), and vegetation dynamics (Forkel et al., 2016). A minor temperature increase in areas with seasonal snow cover can cause changes to the hydrological cycle (Mioduszewski et al., 2014), changes in mass and energy cycles at different scales, a phenology shift (Jeong et al., 2011; Shen et al., 2014), and a prolonged growing season (Blume-Werry et al., 2016; Delbart et al., 2006; Pau et al., 2011). In the Northern Hemisphere, snowmelt infiltration controls surface and subsurface hydrological processes (Ireson et al., 2013), refills the soil water storage, provides water for root water uptake (Nehemy et al., 2022b; Sutinen et al., 2009b), and has a major role in groundwater recharge (Hyman-Rabeler and Loheide, 2023). The timing of snowmelt infiltration also limits the length of the growing season (Vaganov et al., 1999), while the total amount of plant-available water in soil storage can be limited by the interplay of meltwater amount and soil characteristics (Muhic et al., 2023; Smith et al., 2011).

Due to challenging working conditions, understudied cryogenic fractionation processes (Beria et al., 2018; Evans et al., 2016), the constantly reducing number of research sites (Laudon et al., 2017) and an overall lack of field observations (Ala-Aho et al., 2021), studies that implement stable isotopes of water have been less frequent at high latitudes (Tetzlaff et al., 2015). The effect of high water availability during the period of low radiation forcing on the seasonal variability of stem water in sub-arctic forests, which often occurs during the snowmelt season, is still underexplored. Still, the strong contrast between isotopic signals of enriched summer and depleted winter precipitation in sub-arctic areas (Dansgaard, 1964; Rozanski et al., 2013) and the relative isotopic depletion of accumulated snowpack and snowmelt compared to the rest of the hydrological system can serve as a potent tool for ecohydrological research. The substantial

contrasts between snowmelt, typically observed isotopic values at the soil–vegetation interface during the summer season and artificially enriched water are exploited in this study to create a double-irrigation experiment. We used water labelled with deuterium to create a large, isotopically enriched infiltration event and tracked water fluxes of different mobility at the soil–vegetation interface in order to identify the mechanisms of soil water replenishment. We further monitored the evolution of this isotopically enriched soil water pool during the winter, spring snowmelt and one summer season to ascertain the role of snowmelt in the restoration of soil water storage. The main objective of the study was to identify how sub-arctic forest till soils and vegetation respond to infiltration events of high magnitude and explain the unique role of snowmelt in cold climate settings. The main research questions were:

1. What mechanisms control infiltration patterns in sub-arctic forest till soils?

2. How is the spring snowmelt different from an artificial irrigation event of a similar size in soil water dynamics?

3. How do sub-arctic trees respond to extensive infiltration events?

## 2  Methods and data

### 2.1  Study site

The study site was in Pallas, a $4.42\,\mathrm{km^2}$ headwater catchment within the sub-arctic climatic region of northern Finland (Fig. 1b; Marttila et al., 2021). The catchment elevation ranges from 270 to 360 m a.s.l. In the 2003–2019 period, the mean annual air temperature at Pallas was $0.4\,^{\circ}\mathrm{C}$, and the long-term monthly mean temperature was $-11.3\,^{\circ}\mathrm{C}$ in January and $+13.9\,^{\circ}\mathrm{C}$ in July. Mean annual precipitation in the 2008–2019 period was $647\,\mathrm{mm\,yr^{-1}}$, of which $\sim 42\,\%$ fell as snow. Snowmelt onset varies from year to year but usually occurs in May and early June.

The $5 \times 20\,\mathrm{m}$ experimental plot (EP) (Fig. 1a, b) is located at Kenttärova, a forested hilltop at the highest point in the catchment (67°59.237′ N, 24°14.579′ E, 347 m a.s.l.). The soil at the EP is shallow (less than 2 m depth) and consists of unconsolidated, podzolized tills and sandy tills (Fig. S1 in the Supplement). The underlying granodiorite bedrock is heavily fractured and can be porous up to tens of metres (Johansson et al., 2011).

Soil saturated hydraulic conductivities, estimated using the falling-head permeameter method, range from $1.40 \times 10^{-6}$ to $1.25 \times 10^{-5}\,\mathrm{m\,s^{-1}}$ ($n = 4$) at 5–10 cm depth and from $9.78 \times 10^{-7}$ to $8.92 \times 10^{-6}\,\mathrm{m\,s^{-1}}$ ($n = 3$) at 30–35 cm depth. Deeper soil samples for hydraulic conductivity tests could not be obtained due to the soil compaction and stoniness. Soil dry bulk density and porosity were determined after a pF

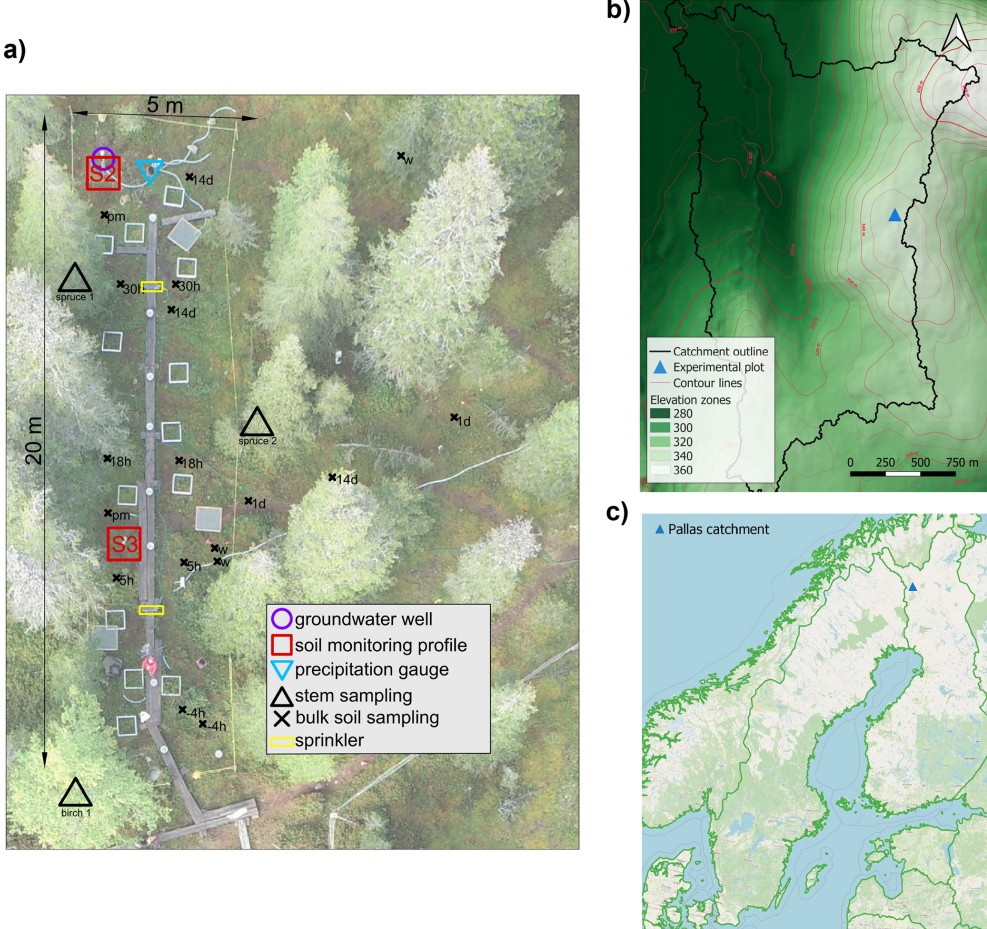

**Figure 1.** Study site overview. **(a)** Sketch of the experimental plot with soil flux monitoring installations. Subscripts next to the bulk soil sampling locations denote sampling times ("w" and "pm" denote the winter 2020 and post-melt 2020 periods, respectively, while the numbers before "h" and "d" denote the time since the start of the experiment in hours and days, respectively). The aerial photograph was taken by Bastian Steinhoff-Knopp (Leibniz University Hannover, September 2018). The grey and white rectangles seen along the boardwalk are soil methane flux measurement collars that were installed prior to our experiment and remained at the study site but did not have a function during the experiment. **(b)** Topographic map of the study catchment. **(c)** Location of the study catchment within northern Europe (blue symbol; 68° N, 24.2° E).

curve test using a 5 bar pressure plate extractor (cat. # 1600, SoilMoisture Equipment Co., Santa Barbara, CA, USA). Mean dry bulk density increases from $1\,\mathrm{g\,cm^{-3}}$ at 5–10 cm to $1.9\,\mathrm{g\,cm^{-3}}$ at 60–65 cm depth, and mean porosity decreases from 0.53 ($n = 4$) to 0.26 ($n = 3$). Mean soil organic matter content, calculated after burning the soil samples at 550 °C in a muffle furnace (Naber N50, Nabertherm), gradually reduces from 10 % at 5–10 cm to 2.65 % at 30–35 cm depth and is less than 1 % below 60 cm depth. Based on soil temperature and frost depth monitoring and previous research by Sutinen et al. (2009a), we can see that soils at Pallas can experience seasonal freezing at a surficial frost depth. However, soil temperatures within the EP stayed above 0.25 °C during the study period (Fig. S2). Mean annual snowpack depth at this location, measured since 2008, was 105 cm (Lohila et al., 2015), but the 2019–2020 winter was unusually snowy and

the maximum observed snowpack depth was 130 cm (Noor et al., 2023). The tree stand consists of Norway spruce (*Picea abies*), with a live canopy that covers almost the entire length of the tree stem, and pubescent birch (*Betula pubescens*) (Aurela et al., 2015).

Two soil profiles (S2 and S3 in Fig. 1a) in the EP were instrumented using time-domain reflectometry soil moisture and temperature sensors (Soil Scout, Soil Scout Oy, Finland) and PTFE/quartz suction lysimeters (Prenart Super Quartz standard, Prenart Equipment ApS, Denmark) at 5, 30, and 60 cm depth. A self-made stainless-steel pan lysimeter of size $20 \times 30$ cm was installed at 35 cm depth and connected via a PTFE tube to a 1 L sample-collecting bottle that was placed at 1 m depth. Additionally, a 1.3 m deep groundwater well with the bottom 40 cm screened and a tipping-bucket precipitation gauge with a GP-HR general-purpose logger (Tru-

Track, UK) were installed next to the S2 profile. Soil property characterization and all field installations other than the precipitation gauge setup were completed in May and June 2018, thus providing a full year for the instruments to equilibrate with the soil, minimizing any structural disturbance before the irrigation experiment commenced.

## 2.2 Irrigation experiment

An irrigation experiment at the EP was conducted in late August 2019 (27 August 16:00 to 29 August 22:00 EEST). Once every 1–2 h, a 1000 L water tank was filled with tap water ($\delta^2$H = −104.36 ‰) spiked with 35 mL of 99.999 % deuterium oxide ($D_2O$) and consequently transported to the site and distributed around the EP using two sprinklers. The sprinkler setup was installed by Määttä (2020) and maintained by Korkiakoski et al. (2022), and sprinklers were positioned such that irrigation water could be distributed evenly within the EP, covering an area of width 3–5.5 m and length 10–21 m (a total area of approximately 118 m$^2$ in calm weather). Weather conditions during the experiment were favourable, with a relatively low wind speed (2.98 m s$^{-1}$ on average, occasional wind gusts of up to 9.6 m s$^{-1}$), a stable wind direction (mostly between 200 and 250°) and no rainfall. The initial plan was to apply $\sim$ 240 mm of water labelled with deuterium, an amount that roughly corresponds to a typical snowmelt event at Kenttärova, based on a 1 m deep snowpack with a snow water equivalent of $\sim$ 240 mm. Water ponding on the soil surface was first observed after 12 h, and the irrigation was stopped once the surface ponding started extending downslope of the EP. The total duration of the irrigation was 30 h, during which twenty 1000 L water tanks of deuterated water were applied to the EP. A total amount of 163.6 mm of irrigation water was recorded by the tipping bucket precipitation gauge, which can be considered representative of the actual conditions as it is only slightly lower than the calculated amount of 169.5–174 mm (considering 20 m$^3$ of water and an area of 115–118 m$^2$). The total difference of 6–10 mm between the measured value and the calculated range roughly corresponds to 35–50 L of water loss per tank, which is reasonable considering that water pump which moved the water from the tank to the sprinklers had to remain submerged at all times, meaning that not all water could be evacuated from the tanks. Furthermore, the irrigation rate was not influenced by wind speed or direction (Fig. S7). The mean (± standard deviation) weighted $\delta^2$H value of the irrigation water was 79.38 (± 5.3) ‰ (Fig. 2a). The temporal variability of the irrigation intensity ranged between 2 and almost 11 mm h$^{-1}$ and led to fully saturated conditions in the EP soil. Although the spatiotemporal distribution of irrigation water was not completely uniform, plot-wide surface water ponding and increased soil moisture showed that the main goal of the experiment, i.e. the simulation of a high-magnitude infiltration event, was achieved.

## 2.3 Data collection

Data collected during the study include soil moisture content, groundwater level, groundwater samples, suction and pan lysimeter water samples, and ponded surface water samples as well as soil core and stem samples.

Soil temperature and volumetric soil moisture were measured every $\sim$ 20 min, and the groundwater level was measured every 10 min using a water level logger (Levelogger, Solinst Canada Ltd., ON, Canada). During the irrigation, a portable vacuum pump and compressor system (Prenart Equipment ApS, Denmark) was used to continuously apply 600 hPa of vacuum to the suction lysimeters, as samples were collected hourly. Groundwater and pan lysimeter samples were sampled concurrently with suction lysimeters. Samples of the ponded surface water were sampled opportunistically. The suction lysimeters were also sampled every 2 weeks during the summer season in 2019 and 2020 and every 2 d, if there was any water available, during the 2020 snowmelt by applying 600 hPa of suction for a period of $\sim$ 4 h. Pan lysimeter water samples for isotope analysis were collected simultaneously with a hand-operated vacuum pump, while groundwater was occasionally sampled with a bailer.

Soil cores from the EP were collected four times during the experiment, using a percussion drill (Cobra 148, Atlas Copco) with a window sampling tube extension (RKS with a reinforced cutting edge with a core cutter; Ø80 mm × 1 m; GEOLAB Paweł Szkurłat). Two replicate cores were collected each time, and all cores were sampled at 5 cm increments down to 50 cm soil depth and at 10 cm increments from 50 to 100 cm depth. Furthermore, soil coring campaigns monitoring the seasonal changes in the EP were conducted 2 weeks after the experiment, at the peak snowpack in April 2020 and after the 2020 snowmelt in mid-June. Soil cores immediately downslope of the EP were collected 1 d after the experiment, again 2 weeks later and under the deep snowpack in April 2020. Stem samples of three trees located in the EP (labelled birch 1 and spruce 1 and 2 in Fig. 1a) at a height of $\sim$ 2 m were collected each day of the experiment, five more times over the next 20 d and once more after the 2020 snowmelt. More details about the samples obtained from the soil core sections (from here on referred to as bulk soil water) and stem sample handling in the field are provided in Muhic et al. (2023).

We define the bulk soil water in line with other isotope-related soil water works such as Geris et al. (2017) as the water extracted from the soil that represents a mix of all waters stored in the soil, ranging from soil matrix water to highly mobile water. Suction lysimeters are assumed to sample soil waters of lower mobility than the water moving through macropores but of higher mobility than the soil matrix water. The difference in isotopic signal between bulk soil samples and suction lysimeter samples represents a combination of tightly bound soil matrix water and macropore wa-

ter isotopic signals, depending on the soil moisture content. The isotopic signal of pan lysimeter water is assumed to be the most realistic representation of highly mobile soil water. Bulk stem water is considered to reflect a mixture of various stem water pools that contain waters of different ages.

## 2.4 Additional data

Supplementary isotopic data used in the study include stable water isotope ratios ($^2$H) of rainfall, snowmelt and bulk soil. Event-based rainfall samples were collected manually and using an automatic ISCO sampler (model 6712, Teledyne, NE, USA) located 2 km northwest from the EP. Snowmelt samples from a snowmelt lysimeter situated within 100 m of the EP were collected daily throughout the melting periods in 2019 and 2020 (Noor et al., 2023). Bulk soil water samples were collected from a nearby ($<100$ m) forested plot (location SF1 in Muhic et al., 2023) on three occasions after the experiment and were used to assess the difference between irrigated and natural bulk soil water isotope compositions. Additional hydrometric data comprised precipitation amount, air humidity, surface air temperature and net radiation data at 10 min intervals – all of which were obtained from an eddy flux tower located immediately next to the EP – and snowmelt flux data from the aforementioned snowmelt lysimeter. The air vapour pressure deficit was calculated using the hourly air temperature and humidity using the formula from Foken (2008).

## 2.5 Isotope analysis

The direct equilibration method (DVE), as described in Wassenaar et al. (2008), was used to measure the isotopic composition of bulk soil water using a cavity ring-down spectroscopy (CRDS) analyser (model L2140-i, Picarro Inc., Santa Clara, CA, USA) located at the University of Oulu. The mean standard measurement deviations for the $\delta^{18}$O and $\delta^2$H values were 0.114‰ and 0.284‰, respectively. We refer to Muhic et al. (2023) for a detailed description of the analysis procedure. Isotopic signatures of liquid water samples from the groundwater well and suction and pan lysimeters were analysed using the same Picarro analyser combined with an autosampler and a vaporizer unit.

Tree stem water was extracted by a cryogenic vacuum distillation (CVD) method at the University of Utah Biology Department (Salt Lake City, UT, USA) following the protocol described by West et al. (2006) and Bowling et al. (2017). The extraction was facilitated using a hot bath filled with water at 100 °C, and extraction times ranged between 60 and 90 min. The extracted stem water was subsequently treated with activated charcoal for 48 h. The isotopic composition ($\delta^{18}$O and $\delta^2$H) of the stem water was measured on a Picarro L2130-i CRDS analyser, and ChemCorrect software was used to examine possible sample contamination.

The isotopic compositions of all samples were determined relative to the Vienna Standard Mean Ocean Water international standard and are shown in $\delta$ notation according to Craig (1961) and Gonfiantini (1978).

## 3 Results

### 3.1 Soil water fluxes during the irrigation experiment

The $\delta^2$H values of water fluxes at the EP observed before the irrigation ($-4$ h), during the irrigation (0–30 h), and during the 3 d immediately after the irrigation was finished (30–89 h) are shown in Fig. 2. We refer to the 0–30 h period as the irrigation period and to the 0–89 h period as the experiment.

The groundwater level (Fig. 2b) was 110 cm below the surface at the start of the experiment and started to rise rapidly after 8 h. The rate of groundwater level rise went down after reaching a depth of $\sim 35$ cm, but the level generally continued to rise until 4 h after the end of irrigation, reaching up to 11 cm below the ground surface. The groundwater $\delta^2$H signal was strongly correlated with both the groundwater level (Mann–Kendall $t = -0.83$, $p < 0.01$) and the total amount of irrigation water applied ($t = 0.92$, $p < 0.01$).

At the onset of irrigation, the isotope ratio of water sampled from the suction lysimeter showed a gradual change from an isotopically more enriched signal in the upper soil (green lines in Fig. 2c) to a depleted signal at the deepest measurement points (black lines in Fig. 2c). The suction lysimeter samples were in a similar isotopic range to the bulk soil water samples (vertical orange lines in Fig. 2c). During the irrigation, the isotopic signals in the two monitored profiles differed greatly, especially at 5 and 30 cm depths. Ten hours after the start of the experiment, the $\delta^2$H signals from the suction lysimeters in profile 3 at 5 cm and profile 2 at 30 cm started becoming more enriched relatively quickly compared to the other locations. On the other hand, the suction lysimeters in profile 2 at 5 cm and in profile 3 at 30 cm showed no significant isotopic changes despite the full saturation of the EP, as was evident from surface water ponding and the high groundwater level. The fastest isotopic response in the EP was observed at a 60 cm depth (dashed black line in Fig. 2c) after only 5 h. Soil moisture and isotope dynamics were well synchronized at all sampling locations during the experiment, as the increase in soil moisture was followed by the enrichment of the water sampled using the suction lysimeter in deuterium. These paired dynamics were decoupled after irrigation stopped. Although more than 160 mm of water was applied, the soil moisture in some relatively shallow layers (profile 2 at 5 cm and profile 3 at 30 cm depth) did not reach saturation values.

The EP was used in another irrigation study in 2018 and 2019, with an aim to increase the overall soil water content at the plot (Korkiakoski et al., 2022). Thus, it was frequently irrigated using tap water with a constant isotopic signal. As

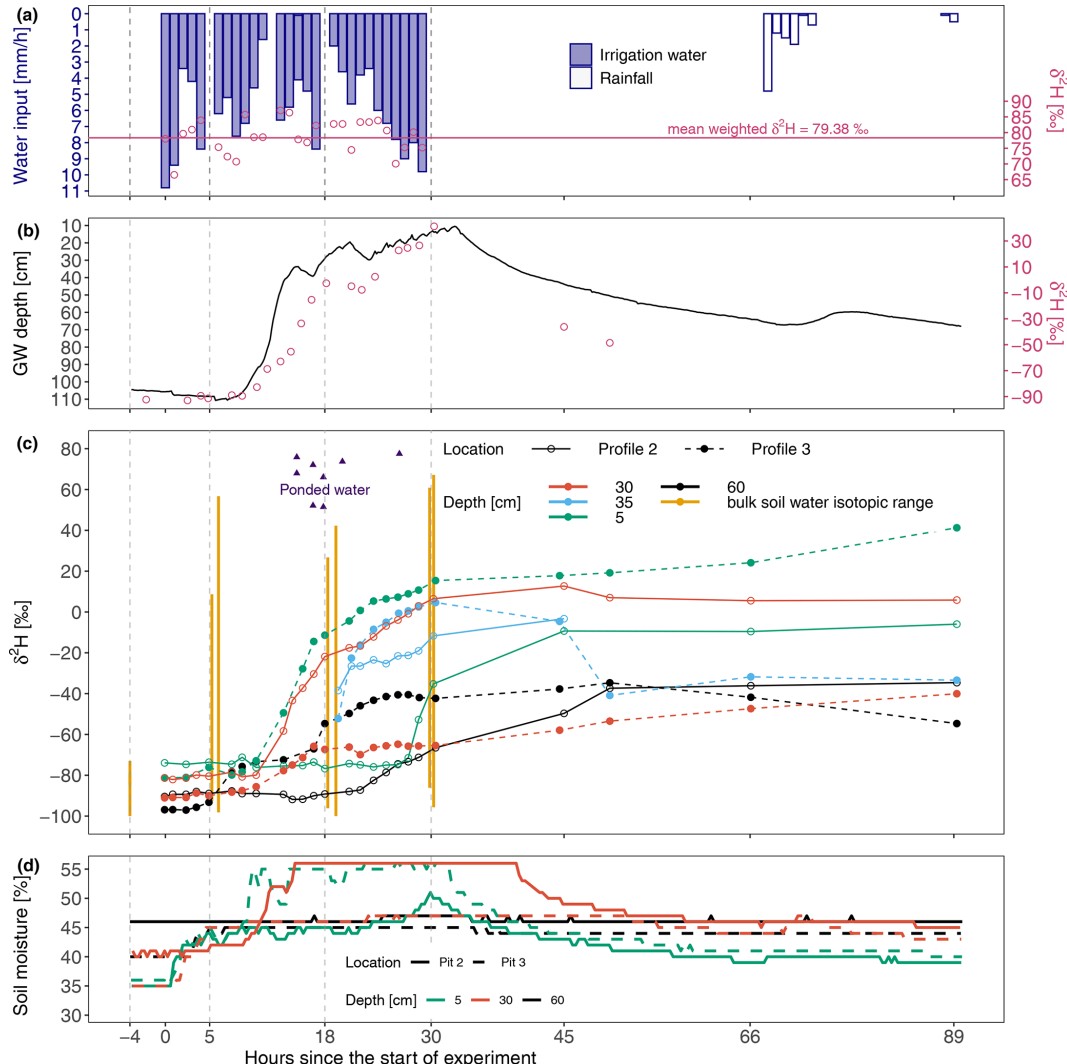

**Figure 2.** Soil water dynamics during the irrigation experiment. **(a)** Hourly water input intensity (bars). The dots show $\delta^2$H values of irrigation water, and the vertical dashed grey lines indicate the timing of bulk soil sampling. **(b)** Groundwater table dynamics; dots represent the $\delta^2$H values of groundwater. **(c)** Soil water $\delta^2$H signal. Suction lysimeters are located at 5, 30, and 60 cm depth. The blue lines and dots (35 cm depth) show pan lysimeter data, while the vertical orange lines denote the bulk soil $\delta^2$H variability. **(d)** Volumetric soil moisture.

the study ended only several weeks before the onset of this study, bulk soil water $\delta^2$H depth profile variability (Fig. 3) was relatively low at the start of the experiment. Topsoil isotopic values were enriched quickly, as a substantial change in the top 20 cm of soil was observed in the first samples collected 5 h after the start of irrigation. Isotopic profiles then barely changed between 5 and 18 h despite the application of ∼ 60 mm more irrigation water. In fact, the signal at 10–20 cm depth became more depleted. Still, at the end of irrigation, at the 30 h mark, the bulk soil water $\delta^2$H signal in the top 15 cm was strongly enriched, and traces of deuterated water could be observed down to 45 cm depth. The bulk soil water in the upper layers was constantly more enriched than the soil lysimeter water, even compared to the lysimeter samples taken up to 2 h after soil coring. In fact, the bulk soil

water $\delta^2$H signal collected at the 5 h mark was more enriched than any soil lysimeter $\delta^2$H signal detected during the whole observation period. The opposite dynamics were present in deeper (below 30 cm) soil layers, where lysimeter waters showed a more enriched signal. The pan lysimeters were empty at the beginning of the experiment, indicating that no freely draining water had reached the pan-lysimeter collecting bottles in the period prior to the experiment. Furthermore, no water was found in the pan-lysimeter collecting bottles during the first 20 h of the irrigation, although approximately 100 mm of irrigation water was applied to the plot during that period. The samples only appeared after the groundwater level exceeded the installation depth of the pan lysimeters (35 cm) and the collecting bottles were "overtopped" by the rising groundwater. From that point onwards, large quantities

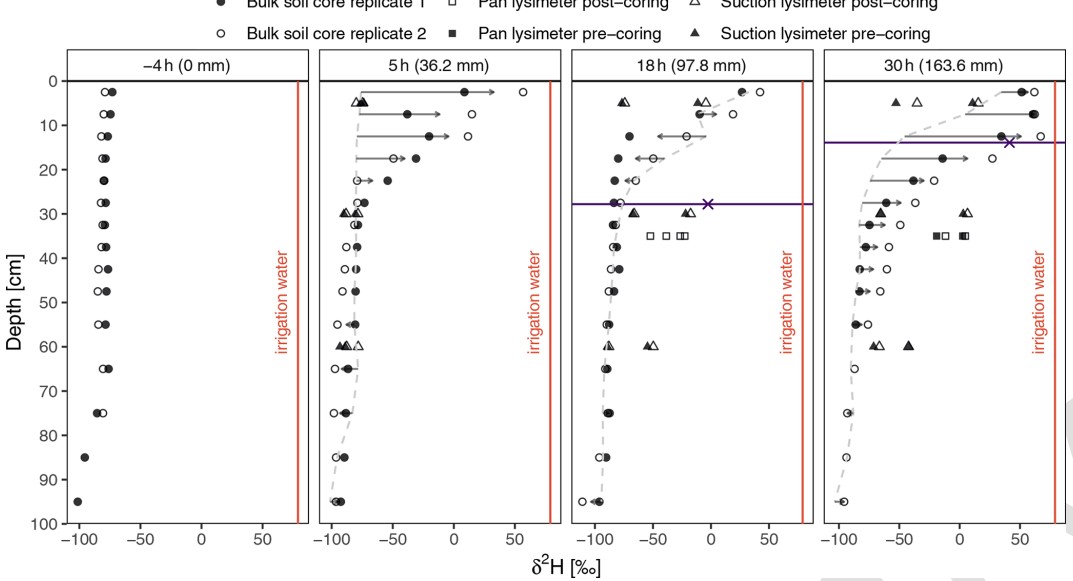

**Figure 3.** Bulk soil water $\delta^2$H dynamics during the experiment (dots). Triangles denote soil lysimeter $\delta^2$H values collected within 2 h before (pre-coring) and after (post-coring) bulk soil sample collection. The differences between the mean values of current bulk soil isotopic profiles and previously sampled (dashed grey line) isotopic profiles and the directions of isotopic changes are indicated by arrows. The horizontal line shows the groundwater level and the groundwater isotopic composition is marked on the line.

of water were evacuated from the collecting bottles each hour and the $\delta^2$H value of the pan lysimeter water rose sharply from $-52$‰ to $2.5$‰, reaching its most enriched values at the end of the irrigation (at 30 h). The isotopic signal got more depleted immediately afterwards, thus displaying similar dynamics to the groundwater isotopic signal. Ponded water samples were more enriched than the groundwater or soil water and closely resembled the labelled water signal.

### 3.2 Changes in soil water fluxes and isotope values after the experiment

The 2020 winter season was characterized by a distinctively deep snowpack (Noor et al., 2023) and a subsequent snowmelt event of above-average size ($\sim 350$ mm) (Fig. 4a). During the snowmelt, the groundwater level in the EP stayed close (up to 10 cm) to the surface for $\sim 2$ weeks. After this, it displayed a quick drawdown followed by another short-lasting peak (Fig. 4b). Soil moisture levels remained relatively high throughout the observation period, and there was a notable soil moisture increase at all depths and in all profiles during the snowmelt event.

The first soil lysimeter water samples, collected at the onset of snowmelt, were still strongly enriched compared to the values that are naturally observed in Pallas soils (Muhic et al., 2023) because a portion of the irrigation water remained in the soil over the winter (Fig. 4d). The most enriched values were observed in deeper soil, while the topsoil isotopic composition was strongly depleted. The enriched water was gradually and fully removed during the snowmelt (see the

zoomed area in Fig. 4d), as the isotopic variability of soil waters sampled using suction lysimeters continuously reduced and converged towards snowmelt values. Consequently, the soil water isotope dynamics for the rest of the summer resembled the previous summer's dynamics (Fig. S3), as the $\delta^2$H values were steadily enriched through the summer, with deeper lysimeters showing more depleted signals than the lysimeters in the upper soil. Pan lysimeter samples were only available during the snowmelt, as the collecting bottles were empty throughout the summer season, and a progressive depletion dynamic similar to that seen with deeper suction lysimeter samples was observed.

The effect of irrigation on the bulk soil water isotopic signal was assessed by comparing the observed $\delta^2$H values in the EP, a nearby reference plot not affected by irrigation and an area downslope of the EP (Fig. 5). In the EP, autumn rainfalls fully shifted the strongly enriched signal in the upper ($<10$ cm) soil layers towards the values observed in the reference plot (green lines in Fig. 5). The deuterium peak moved down to a depth of 32.5 cm and enriched values were also observed at a depth of 95 cm (second panel in Fig. 5a). The depth distributions of $\delta^2$H in two soil cores from the EP were similar, but $\delta^2$H values were more enriched in the EP replicate 1 (Fig. 5a). The $\delta^2$H depth dynamics in soil cores from the winter period showed more of a resemblance to the cores obtained in the non-irrigated plot, with comparable values in the upper 40 cm, while the peak dropped to 90–100 cm. After the snowmelt, the $\delta^2$H signals in the EP were as depleted as the signal in the reference plot, with near-identical signals detected at all four cores.

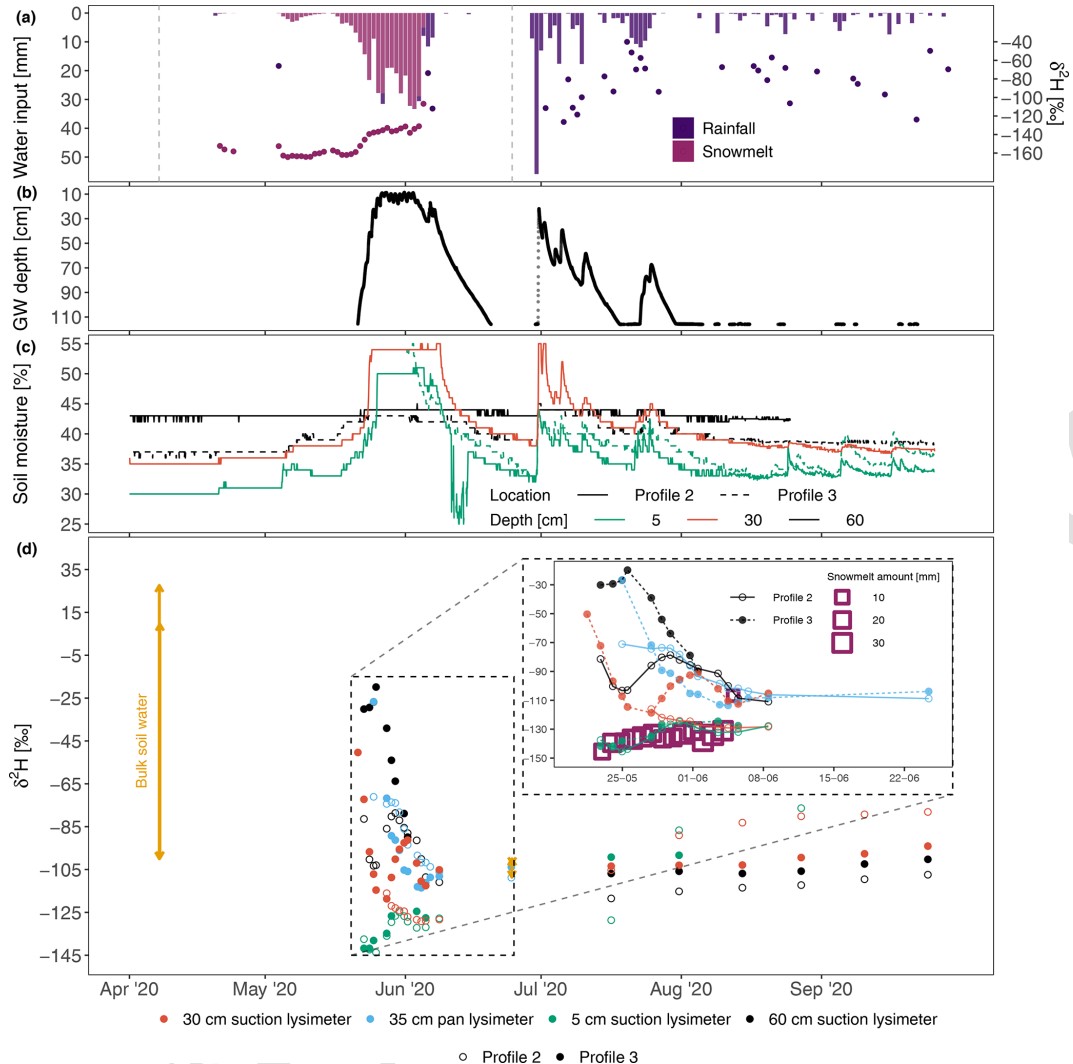

**Figure 4.** Water fluxes at the irrigation plot during the whole observation period. **(a)** Infiltration event magnitude (bars) and $\delta^2$H signal (dots). The dashed vertical grey lines show soil coring campaign timings. **(b)** Groundwater table dynamics. **(c)** Volumetric soil moisture. **(d)** Soil water $\delta^2$H signal. The range of bulk soil $\delta^2$H values is indicated using purple arrows. The inset graph displays the soil lysimeter water and snowmelt water isotope dynamics during the snowmelt period.

The lateral fluxes which resulted from the irrigation experiment are displayed in Fig. 5b. Two soil cores were excavated 1 d after irrigation ended (see Fig. 1): one in the zone affected by surface water ponding immediately next to the EP (white filled rectangles in the first panel in Fig. 5b) and the other 10 m downslope (black filled rectangles in the first panel in Fig. 5b). The effect of surface water ponding can be clearly identified in the upper 10 cm of the soil, but the depth dynamics of deuterium below 10 cm depth were remarkably similar, and a small peak ($\sim -17\,\permil\,\delta^2$H) was present at 20 cm depth. As in the EP, autumn rainfall shifted the $\delta^2$H signal in the upper soil towards the natural range (14 d after irrigation), but the deuterium breakthrough curve peak moved downwards at a faster rate than in the EP. In winter, soils in the EP and downslope of the EP showed similar $\delta^2$H depth dynamics,

but the signal in deeper soil layers (below 60 cm) was more enriched in the EP.

## 3.3 Bulk stem water $\delta^2$H dynamics

The trees for stem water sampling were selected in 2018, but the irrigated zone was moved slightly in 2019, leaving the birch tree 30–40 cm outside of the irrigation zone (Fig. 1a). Nevertheless, it can be assumed that the lateral spread of the birch tree's roots is large enough to reach the irrigation zone. The $\delta^2$H values of samples obtained before the irrigation event were clustered around $-75\,\permil$ ($-74.61\,\permil$, $-72.70\,\permil$ and $-78.00\,\permil$), but the stem water dynamics of birch and spruce trees were different during the experiment. The birch tree did not respond strongly to the irrigation event, as the

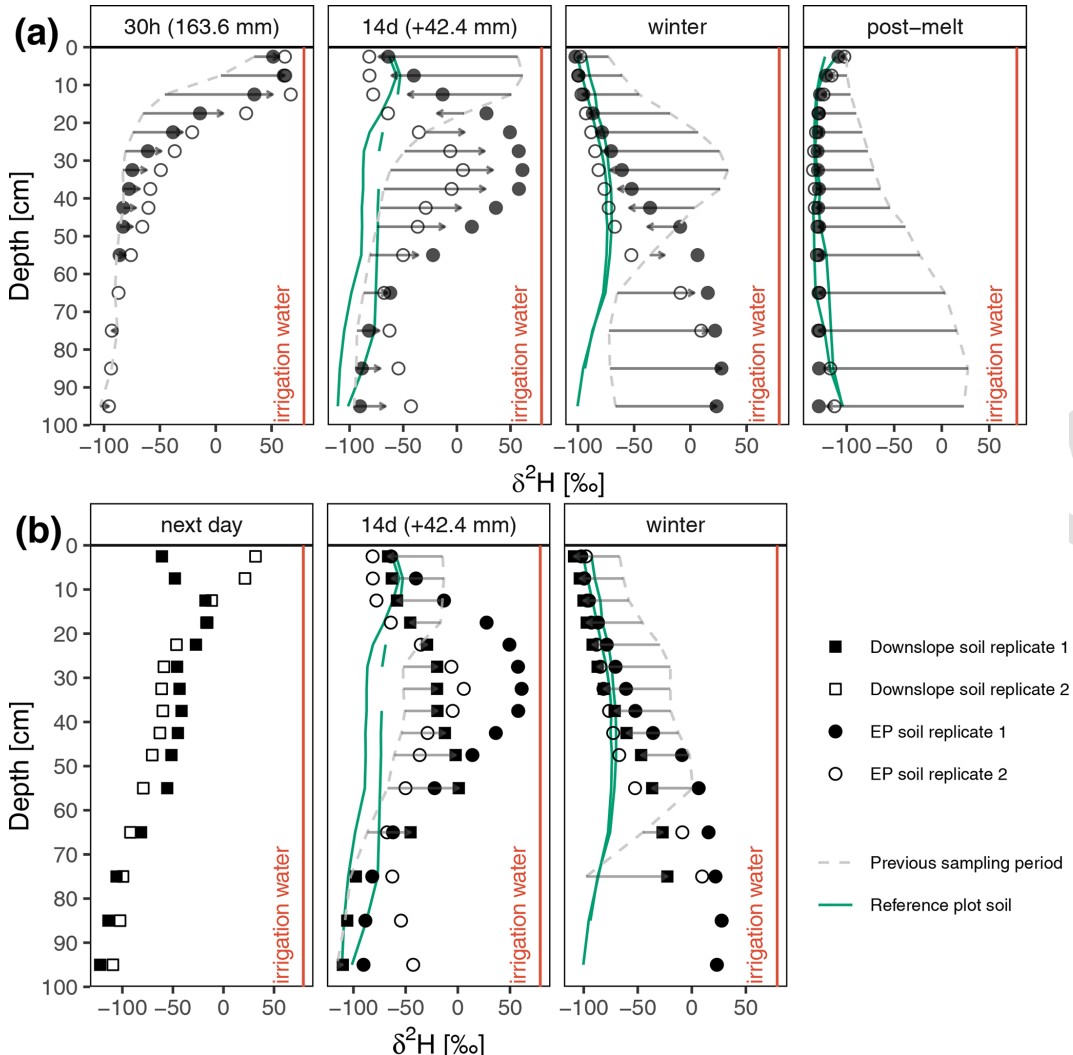

**Figure 5.** Bulk soil water $\delta^2$H dynamics after the experiment in **(a)** the irrigation plot and **(b)** soil immediately downslope of the EP. Green lines show the bulk soil water dynamics in a nearby reference plot that were collected at the same time. The isotopic change between the mean values of current isotopic profiles (dots) and the previously sampled (dashed line) mean isotopic profile in the EP are indicated using arrows. Bulk soil $\delta^2$H values collected from the EP in the same period are displayed for comparison.

stem water isotopic signal remained mostly stable throughout 2019, while the spruce stem water dynamics were characterized by two peaks: a smaller one ($\Delta\delta^2$H of $+4\,‰$ and $+12\,‰$) at the end of irrigation (Fig. 6b, day 1) and a larger one after 15 d ($\Delta\delta^2$H up to $+80\,‰$ compared to the pre-irrigation signal). In the summer of 2020 (day 302 after the start of the experiment), all samples displayed a strongly depleted $\delta^2$H signal again ($\sim -112\,‰$), comparable to the $\delta^2$H signal observed in both bulk and lysimeter soil water.

## 4 Discussion

We simulated an infiltration event similar to snowmelt on an experimental plot with shallow till soil located on a forested hilltop in a sub-arctic zone by applying 160 mm of deuter-

ated water over a 30 h period. The resulting soil water fluxes displayed high spatiotemporal variability, identifiable in both hydrometric and isotopic data. We further observed a fast groundwater table rise, bypass flow, surface water ponding and a fast isotopic response in stem water during the irrigation. An isotopically enriched signal was preserved in the soil storage throughout the winter period and then got fully flushed out by the spring snowmelt, leading to strong isotopic homogenization of the soil. Furthermore, the homogenization was not limited to the soil but extended throughout the soil–vegetation interface. Soil lysimeter waters at all depths gradually shifted towards more positive values over the following summer season, forming a pattern of enrichment reduction with depth. Observed soil water fluxes at the soil–vegetation interface uncovered infiltration mechanisms

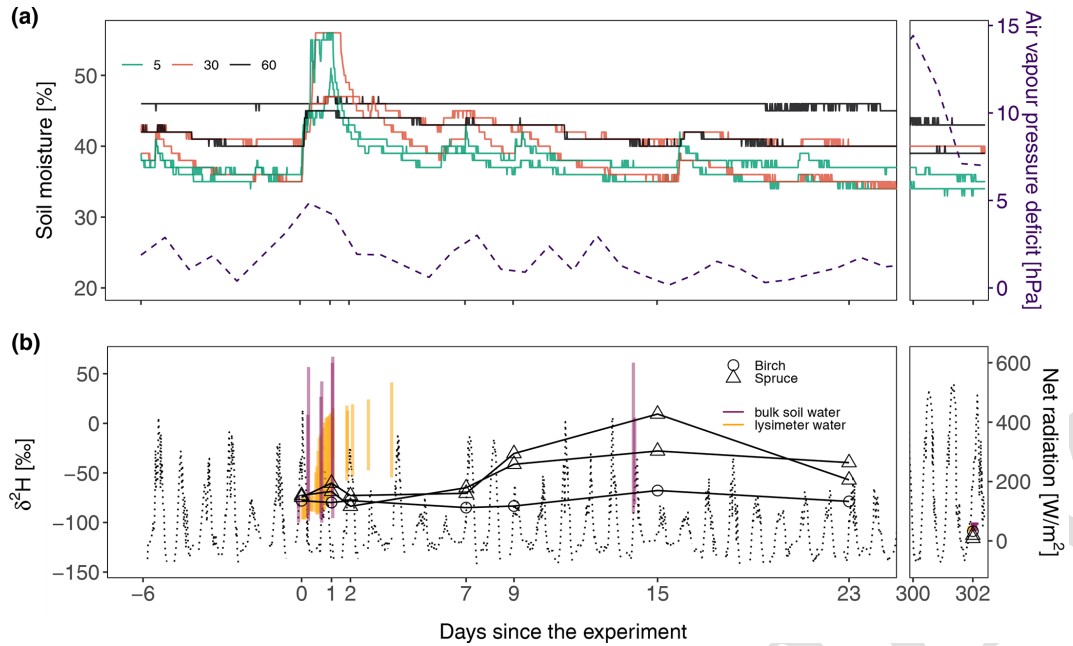

**Figure 6.** Water fluxes at the soil–vegetation interface. **(a)** Volumetric soil moisture (solid lines) and air vapour pressure deficit (dashed line) dynamics. **(b)** The bulk stem water isotopic signal (solid lines and triangles) and net radiation (dashed lines). The vertical orange lines show the $\delta^2H$ variability of soil lysimeter water, while the vertical purple lines show the $\delta^2H$ variability of bulk soil water. The ggbreak package (Xu et al., 2021) was utilized for making this graph.

in shallow arctic till soils and highlighted the unique role of snowmelt in soil water replenishment. We concur that a major fraction of the infiltrating water moves through the macropore network and bypasses the deeper soil matrix, meaning that the groundwater level rise triggered by the infiltration excess also largely occurs within the macropore network and gradually increases the exchange between the macropores and soil matrix. Three infiltration stages were detected (Fig. 7):

1. First stage – the initiation of macropore flow. The surface microtopography induces surface ponding and focused infiltration. Infiltrating waters first fill the surficial soil matrix but, due to the limited infiltration and water-holding capacity of the surficial soil matrix, the water fluxes are conveyed downwards through the macropore network. The fluxes are largely vertical and the macropore flow is unsaturated. The soil matrix in deeper soil layers is bypassed.

2. Second stage – development of lateral fluxes. As the transport capacity of certain parts of the macropore network in deeper soil is surpassed and infiltrating water percolates towards the soil layers with lower permeability, groundwater rises towards the surface through the macropore network. The buildup of pressure in the saturated macropores leads to increased horizontal hydrologic connectivity in the macropore network, while groundwater advancement to more conductive upper

soil layers initiates subsurface lateral flows through a transmissivity feedback mechanism. Increased surface ponding generates surface flow via a fill-and-spill mechanism. Isolated and air-filled macropores, as well as a substantial part of the deeper soil matrix, are still largely unaffected by the infiltration.

3. Third stage – soil matrix refilling. The constant water supply combined with a prolonged period with a high groundwater table leads to increased hydrological connectivity within the macropore network at all depths. Water diffusion into textural pores and pore water exchange between macropores and the soil matrix are bolstered due to the increased contact surface and increased time for dispersive exchange, thus promoting soil water homogenization and soil matrix flow and further increasing soil wetness and hydraulic conductivity. Shorter distances between water-filled macropores, combined with near-saturation hydraulic conductivity allow matrix flow to reach all soil pores.

Each infiltration stage, as well as the effect of the observed infiltration mechanism on the vegetation in seasonally snow-covered areas, will be discussed in the following sections.

### 4.1 The initiation of macropore flow

Isotopic heterogeneity of soil waters is universally observed, and it would be advantageous to identify under what conditions isotopic homogeneity (Sprenger and Allen, 2020) and

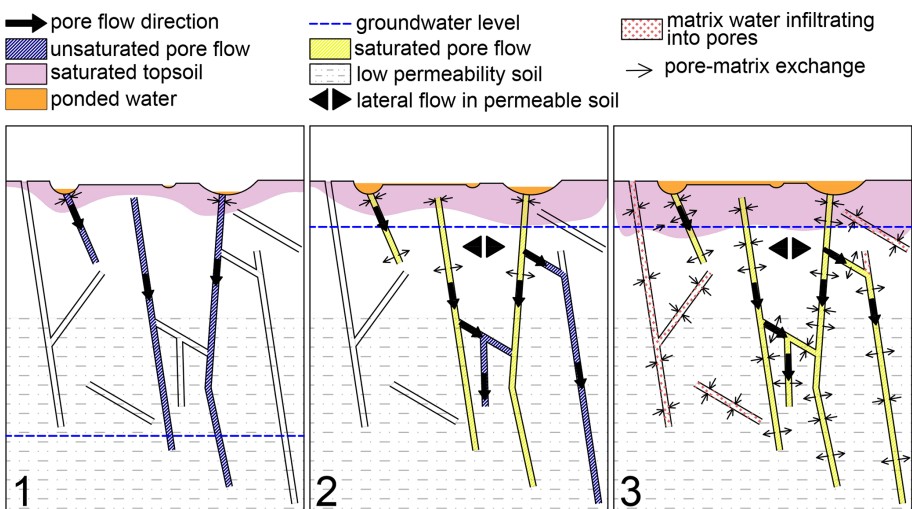

**Figure 7.** Infiltration mechanisms in shallow sub-arctic till soil: (1) the initiation of macropore flow; (2) the development of lateral fluxes; and (3) the refilling of the soil matrix.

full soil matrix water displacement can be achieved. The amount of water applied to the soil during this experiment was relatively large considering the shallow soil at this location, yet soil water storage continuously displayed a strong heterogeneity in soil moisture and isotopic measurements in the two observed profiles. Our data show that some soil patches at 5 cm depth remained largely isolated from infiltrating waters for a long period (solid green lines in Fig. 2c and Fig. 2d), despite the occurrence of isotopic changes in deeper soil layers at the same profile (solid red lines in Fig. 2c and d) and plot-wide surface water ponding. Such an agreement between isotopic and hydrometric responses was also found by Laudon et al. (2004) in a study conducted in till soils overlaid by organic soil layers of different depths. The immediate increase in soil moisture and strongly enriched $\delta^2$H signal in the upper layers of bulk soil samples collected after 5 h with no change in the isotopic composition of suction lysimeter water (Figs. 2c and 3) suggest that, in agreement with the principles of soil physics, the slow domain (smaller pores in the soil matrix) is filled first (Hrachowitz et al., 2013; Hutson and Wagenet, 1995). During the first 5 h of the irrigation, the average soil moisture content in the top 10 cm increased by 7 percentage points (from 35.5 % to 42.5 %), while the average bulk soil $\delta^2$H value in the top 10 cm increased by 88 ‰, from −77 ‰ to +11 ‰. During the same period, 36.2 mm of irrigation water with a $\delta^2$H value of 76.9 ‰ was applied to the plot. Assuming that (1) all additional water in the topsoil is represented by the soil moisture content increase and (2) the bulk soil $\delta^2$H signal only changes due to the simple mixing of the antecedent and newly infiltrated water, these additional 7 percentage points of soil moisture would have to have a $\delta^2$H signal in the range of +460 ‰ to shift the bulk soil water $\delta$2H signal from −77 ‰ to +11 ‰. Alternatively, if the $\delta^2$H value of the irrigation water in the observed 5 h pe-

riod (76.9 ‰) was considered as the endmember of the newly infiltrated water, a soil moisture increase of some 41 percentage points would be required to result in a bulk soil water enrichment of +88 ‰. Following this, it is clear that the assumptions of well-mixed conditions and piston flow are both simplifying the processes at play and that there are some limitations to using soil moisture content as a sole indicator of bulk soil $\delta^2$H signal variability. Namely, the soil moisture content cannot accurately represent soil waters of all mobilities, especially in the case of fast draining and macropore water, and it furthermore cannot indicate the soil water displacement process. The only way to actualize such a strong shift in the bulk soil $\delta^2$H signal is through a combination of two processes: (1) partial soil matrix water displacement and (2) mixing with highly mobile infiltrating or macropore water. Furthermore, neither of these two processes are clearly visible in the suction lysimeters' $\delta^2$H values, as they do not sample waters with very low mobility (soil matrix water contained in the smaller pores) and generally do not show an immediate response to more mobile or macropore waters. Brooks et al. (2010) observed that infiltrating water first enters smaller pores, but the filling of smaller pores subsequently causes a decrease in soil matric potential, so the infiltrating water starts bypassing the soil matrix and moving through bigger pores and preferential flow pathways. Column studies under near-saturated conditions have further underlined the role of well-connected networks of smaller pores in preventing the initiation of macropore flow (Larsbo and Jarvis, 2005). In their dynamic partial mixing model, Hrachowitz et al. (2013) concluded that the relocation of flow to larger pores limits dispersive exchange between preferential flow pathways and the soil matrix by reducing the contact surface and contact times between these soil domains. The routing of water to preferential flow pathways can be addi-

tionally exacerbated under near-ponding conditions (Bogner et al., 2013; Kulli et al., 2003; Li et al., 2013), such as the ones observed in this study (Fig. S4), due to the limited soil infiltration capacity. The transfer of flow towards the macropore network is further increased by the low water-holding capacity of surficial soil at Pallas (Fig. S6). The reduced interaction between macropores and the soil matrix may be one possible explanation for the offset between soil bulk and lysimeter waters sampled at the same time and depth (Fig. 3). Studies which have focused on the isotopic difference between bulk and lysimeter waters have often had inconclusive results regarding the main drivers of this separation (Finkenbiner et al., 2022; Oshun et al., 2016; Sprenger et al., 2018a; Xiao et al., 2020), but they agree that the interaction between macropores and the soil matrix depends on soil characteristics (Scherrer et al., 2007). Isotopic offsets between bulk soil and soil lysimeter water have been estimated in previous studies using an empirical formula (Chen et al., 2016) and via an isotope mass balance approach (Sprenger et al., 2019), but strongly heterogeneous conditions in the EP prevent the application of such formulas.

The infiltration capacity of soils is controlled by their soil texture, structure, antecedent moisture, microtopography and various site-specific processes (e.g. soil freezing), so different infiltration mechanisms and responses to excess infiltration are observed in different soils. Small-scale variation of terrain and surface vegetation can cause focused infiltration and surface ponding, as observed during this experiment (Fig. S3). Small topographic depressions, varying in diameter from centimetres to metres, are often described as ground surface roughness (Hansen, 2000) and strongly impact surface connectivity, storage, overland flow (Darboux et al., 2002; Frei et al., 2010) and soil profile development (Manning et al., 2001). These factors that contribute to the heterogeneity explain how the observed surface water ponding could occur in isolated patches of soil even though the irrigation rate was lower than the measured infiltration capacity of surficial soils. They are commonly observed in many sub-arctic landscapes (Hayashi et al., 2003), where they serve as temporary accumulation spots for meltwater and promote groundwater recharge and bypass flow through focused infiltration (Berthold et al., 2004; Hayashi et al., 2003). Focused infiltration in small ($<1$ m diameter) depressions was also identified by French and Binley (2004). The highly enriched $\delta^2$H signal of ponded water samples (Fig. 2c) collected in the EP suggests that irrigation water was temporarily stored in these small depressions as an infiltration excess due to an inability of the water to infiltrate into the deeper soil layers. As the irrigation progressed, and over 60 mm of water was applied to the EP, the movement of water through the soil was limited not only by the infiltration capacity of the surficial soil but by the infiltration capacity of deeper soil layers too. As previously infiltrated water reached deeper, strongly compacted soil layers, its further percolation was halted, and subsequent infiltrating water started accumulating in the up-

per soil layers, eventually exfiltrating to the surface. In such periods of high soil saturation, preferential flow is usually intensified, as the water is increasingly transported through a small fraction of the total soil pore space (Jarvis, 2007), thus reducing its residence time in the system (Šimůnek et al., 2003). The occurrence of preferential flow is common in glacial till soils (Etana et al., 2013), especially at hilltop sites (Liu and Lin, 2015).

## 4.2 Development of lateral fluxes

We detected that the fast groundwater level rise (Fig. 2b) was strongly correlated with a $\delta^2$H increase in groundwater. Derby and Knighton (2001) described similar processes and detected the formation of a groundwater mound and the swift transport of applied tracers beneath soil depressions. The strong correlation between groundwater level and isotopic signal illustrates how irrigation water mixes with the existing soil water storage; the amount of mixing is related to the distance from the soil surface to the groundwater table. If the rate of groundwater response resulted from Darcian matrix flow, based on the saturated hydraulic conductivity of the upper soil layers (mean value $\sim 5.70 \times 10^{-6}$ m s$^{-1}$), and under a 1 m hydraulic gradient due to elevation, the rate of groundwater level increase would be $\sim 0.5$ m d$^{-1}$. This is an overestimation, because the unsaturated conductivity is lower than the saturated conductivity and soils below 50 cm depth are significantly more compacted and less porous, so their conductivity can be several orders of magnitude lower. Furthermore, the deuterium peak in the depth profiles moved from 10 to 30 cm over the 14 d period after the experiment, and it moved from 30 to 85 cm during the following $\sim 6$ months (Fig. 5a, panels 1–3), indicating much slower movement of water through the soil matrix. A much faster groundwater response can occur in nearly saturated soils in the capillary fringe. As the groundwater level in the EP is typically at least 1 m below the ground during the summer and autumn (Fig. 4b), the capillary rise in sandy till soils in the EP should be limited to the soil layers at a similar depth. Despite this, we observed that the groundwater level increased by 77 cm during the 8 h period between hour 6 and hour 14 of the experiment. The observed rate of groundwater rise and isotopic signal enrichment indicate that groundwater moves via bypass flow through the macropore network. In this way, a hydrological connection between infiltrating water and groundwater is established through the macropore network, and a flow-constricting zone in deeper soil layers is largely bypassed. A strong correlation between the total amount of irrigation water applied to the EP and the groundwater isotopic signal shows that the enriched irrigation water reaches groundwater directly through the macropore network. The increase in soil moisture during the soil wetting stage is clearly an important driver and a necessary precondition for the groundwater level rise, based on the fact that they showed similar dynamics during the entire obser-

vation period (Fig. 4b and c). Still, the generally high soil moisture and correspondingly low soil matric potential in the EP helped convey the infiltrating water towards macropores rather than into the soil matrix and limited the pore water exchange between soil matrix and macropores. Although macropore flow is typically unsaturated due to its high transport capacity (Cey and Rudolph, 2009; Jarvis, 2007) and relative hydrophobicity compared to the soil matrix, we can assume that the groundwater rise through the macropore network signals a temporary excess of water in the soil and the near-saturation of vertical macropores, which consequently intensifies the lateral movement of water and strengthens the horizontal hydrological connectivity in the macropore network. While macropore flow is unsaturated, soil water does not enter the interconnected pores that spread laterally from the main macropore (Fig. 7, left panel). As the saturation level increases, the macropore flow becomes more similar to piston flow, and the buildup of pressure instigates the movement of water into the laterally spreading pores (Fig. 7, middle panel). Similarly, the increase in the saturation level in macropores is followed by an increase in pore water exchange between the soil matrix and macropores. Although direct observations of macropore flow were not possible due to the scale and setup of the study, the enrichment of the bulk soil water isotopic signal in the upper layers compared to the isotopic signal of water from suction lysimeters can serve as an indicator that water of higher mobility was present in the soil and moved through the macropore network even during the first hours of the irrigation. The isotopic enrichment observed at 60 cm depth (Fig. 2b) during the irrigation, which occurred before the enrichment in upper soil layers, further shows that preferential flow pathways were active even in the early stages of the experiment. On the other hand, the abrupt appearance and disappearance of large quantities of water in pan lysimeters, as observed during both the irrigation experiment and the snowmelt, can be used to infer lateral flows and the near-saturation of the macropore network.

Once the groundwater table reached the upper soil layers, lateral soil water fluxes greatly increased, which is evident from the reduction in the rate of groundwater level rise and the amount of water extracted from pan lysimeter sample collection bottles ($\sim 0.5\,\mathrm{L\,h^{-1}}$). As the pan lysimeters did not receive any water in periods when the groundwater level was below their installation depth (35 cm; Fig. 2c), it seems that the flow volume of the freely draining water was not sufficient to reach the sampling bottle. Accordingly, the water samples collected from the pan lysimeters during the experiment likely originate from lateral flow rather than from freely draining water. The development of lateral fluxes due to groundwater rise is commonly observed in northern seasonally snow-covered catchments with glacial till soil (Laudon et al., 2004; Stumpp and Hendry, 2012). Also, the basic requirement for the transmissivity feedback mechanism (Kendall et al., 1999) – saturated hydraulic conductivity that increases towards the surface and near-surface

soil horizons – is fulfilled in the EP (Sect. 2.1 and Table S1 in the Supplement). Furthermore, the rate of groundwater table rise decreased as the groundwater level reached more permeable soil despite the irrigation being applied at the same rate, revealing continuous water loss through lateral fluxes in shallow ($\sim 0$–35 cm depth) soil layers. Bulk soil samples collected 14 d after the experiment (Fig. 5b, panel 2) revealed that the deuterium peak reached a greater depth downslope of the EP than in the EP, which could result from lateral transport caused by groundwater mounding (Stumpp and Hendry, 2012). After irrigation, the groundwater isotope signal recovery (depletion) was rapid, as the enriched retreating groundwater again mixed with the pre-event water from the soil storage (Fig. 2b). The groundwater table rise promotes more intense mixing with the water stored in the unsaturated zone (Rouxel et al., 2011), but the short duration of the high water table and its subsequent fast drawdown did not isotopically homogenize the soil. The negligible isotopic change in the deeper (below 45 cm) soil layers (Fig. 3) shows that the water fluxes in these layers are still largely transported through the macropore network. Aboveground water fluxes that developed during the experiment, like the previously described surface water ponds, slowly expanded and merged into an ephemeral surface water network (Fig. S4) following the fill-and-spill mechanism (Appels et al., 2011; Tromp-Van Meerveld and McDonnell, 2006), which can produce biogeochemical hotspots (Frei et al., 2012) on a plot scale and promote surface–subsurface water exchange (Frei et al., 2010).

### 4.3 Soil matrix refilling

The deuterated water signal was fully removed from the topsoil after rainfall events following irrigation (Fig. 5), consistent with the findings of Rothfuss et al. (2015), who identified that the isotopic composition of the surface soil immediately shifts towards the isotopic composition of the infiltrating water. The enriched signal remained deeper in the bulk soil during the winter and slowly travelled downwards, as deeper bulk soil water samples revealed a delayed response to the irrigation event. Traces of enriched water were also observed in lysimeter water at the onset of snowmelt (Fig. 4d) but were gradually depleted during the snowmelt. Bulk soil samples at all depths, apart from the top 5 cm, showed strong depletion after the snowmelt event. Essentially, the aftermath of the uncharacteristically large snowmelt event was the homogenization of all water fluxes at the soil–vegetation interface (Fig. 4d, right panel in Fig. 5b) and a marked decrease in soil water ages, as described in Sprenger et al. (2018b). This contrasts with the findings of Laudon et al. (2004), who identified no isotopic effect of 200 mm of snowmelt in glacial till soil layers below 90 cm depth, possibly due to deeper (10 to 15 m) soils and a much lower groundwater level. Sprenger et al. (2017) found that the peak bulk soil isotopic variability was perceived after massive precipitation events, but the scale and duration of events described in the study were not

comparable to the typical snowmelt at Pallas and were ultimately unable to cause homogeneous wetting of the vadose zone.

The primary reason for such a drastic effect of snowmelt on the soil water storage in the EP is an extended period (more than 2 weeks) with a high groundwater table combined with a constant snowmelt-sourced water supply (Fig. 4a and b). This long-lasting soil saturation enhances the rate of imbibition of water into the smaller pores through an increased wetted area and promotes equilibrium conditions within the soil continuum by increasing short-range connectivity (Schlüter et al., 2012) and reducing water pressure fluctuations. Numerous studies have revealed that large air bubbles may remain trapped in the macropores (Sněhota et al., 2010), even during the gradual saturation of the soil from below (Luo et al., 2008), and they remain non-conductive to water flow. In addition, the macropore network consists of clusters that do not uniformly spread through soil or reach every soil parcel (Jarvis, 2007). This means that a major part of the soil storage can only be replenished via slow soil matrix flow. Prolonged periods with a high groundwater table can further enhance the matrix flow by augmenting the soil hydraulic conductivity (van Genuchten, 1980) and, consequently, the soil matrix flow rates (Fig. 7).

Long-lasting soil saturation could be less frequent in high-latitude areas in the future, as climate model simulations predict a precipitation shift from snowfall to rainfall (Bintanja and Andry, 2017). As demonstrated by the irrigation experiment, even large summer rainstorms are not likely to produce a long soil saturation period due to their higher intensity and shorter duration. Prolonged soil saturation is also less likely to occur on steep slopes, as Mueller et al. (2014) found highly spatially variable soil isotopic signals during the snowmelt period. However, they note that the soil sampling campaign performed in their study took place roughly 4 months after the snowmelt, which is arguably a longer time span than the transit times in the upper 1 m of the soil. In a more recent study conducted in a steep alpine catchment, Michelon et al. (2023) recognized the potential of the snowmelt to strongly flush the entire subsurface system and reset the isotopic values of soil waters. Furthermore, homogenization of soil waters during the snowmelt should be expected in topographical lowlands with a shallow groundwater table, where groundwater–surface water interactions are commonly observed in the sub-arctic (Autio et al., 2023).

Contrary to the observed isotope dynamics, some hydrographic observations made both during and after the experiment, namely the soil moisture measured by the 60 cm deep sensor in profile 2, indicate that the irrigation and snowmelt events did not produce drastic changes in the soil. The measured soil moisture value at this location was always about 45 % regardless of the hydrological conditions (see the full black lines in Figs. 2d and 4c). While such a soil moisture dynamic could have arguably been caused by a sensor malfunction, it could also showcase the existence of soil sections that are largely isolated from the surrounding soil matrix or macropore network due to a localized soil compaction that is either a natural facet of the soil heterogeneity or is artificially created during the sensor installation. While the full extent of this potential decoupling between certain portions of soil matrix cannot be quantified, it is highly unlikely that a few such isolated soil patches could significantly affect the conclusion of the study, i.e. that a general soil isotopic homogenization occurs as the aftermath of snowmelt.

To understand the limits of the conceptual model's applicability, it is important to assess how the described mechanisms would work in the frozen soil conditions that are expected in the sub-arctic catchments. Although soil freezing in the EP was not directly observed during the experimental period, we suggest that the proposed model is theoretically well suited to the frozen-soil conditions. Namely, the main mechanisms of water movement in frozen-soil conditions, focused infiltration and macropore flow, were also key mechanisms observed during the first infiltration stage in this study. The freezing of the upper soil layers is advantageous for macropore flow development and could possibly trigger the onset of macropore flow even faster than in the case of unfrozen soil. One of the requirements for the groundwater rise through the macropores, which defines the second infiltration stage, is that infiltration exceeds the transport capacity of the soil. As soil pores become filled by the ice, their overall volume and consequently soil transport capacity are reduced, again supporting the proposed infiltration mechanism. At the same time, the ice-filled pores can act as flow-restricting zones that limit the hydrological connectivity between different soil patches, especially in the surficial soil layer, where more extensive freezing should be present. The depth of the freezing front could additionally limit the homogenization effect caused by a shallow and long-lasting groundwater table, but this effect should be limited to the surficial soil.

## 4.4 The influence of vegetation on soil water fluxes

Besides soil characteristics and infiltration rates, soil water infiltration patterns and storage are also affected by vegetation, through species-specific hydraulic redistribution and water uptake (Dubbert and Werner, 2019). Plant water fluxes are controlled by the moisture gradient between the atmosphere and soil, and the saturated soil conditions present in northern regions during the snowmelt often induce plant rehydration (Nehemy et al., 2022b) through positive root pressure (Hölttä et al., 2018). Root water uptake is a passive process that follows the water potential gradient, so saturated soils can drive the water into the roots. In such circumstances, internal plant water storage can be a significant outlet for soil water fluxes. We speculate that the timing of the experiment allowed us to first detect the quick and short-lasting stem water enrichment caused by fast soil saturation and then to detect a more gradual enrichment that resulted

from transpiration-related root water uptake. Although the experiment was set in a period with low net radiation and a low vapour pressure deficit (Fig. 6), we discovered an isotopic response in bulk stem water from spruce trees within 30 h of the start of irrigation, most likely caused by the inflow of water from the fully saturated soil. Such a response was not observed in the birch tree that was located slightly outside of the EP, presumably because the majority of its roots were in a non-irrigated zone. Soil saturation during the experiment was of local in character, as the soil moisture in areas immediately upslope of the EP did not change (Fig. S5). The $\delta^2$H increase in the second spruce tree ($+4\permil$) noted during the experiment was rather low and did not necessarily originate from labelled water, as heterogeneous isotopic signals within the tree (Goldsmith et al., 2019; Seeger and Weiler, 2023; Treydte et al., 2021) could produce an offset of a similar size. However, it is also possible that the labelled water signal was diluted after that water mixed with the pre-existing water in the xylem (Aguzzoni et al., 2022). As the bulk stem waters were extracted using a cryogenic vacuum distillation method and analysed using a laser spectrometer, the isotopic values could have also been affected by extraction biases (Diao et al., 2022) and co-extracted organic compounds (Martín-Gómez et al., 2015; Millar et al., 2018), while the true signal of sap flow could have been additionally masked by exchanges with other water compartments within the stem (Barbeta et al., 2022; Fabiani et al., 2022; Nehemy et al., 2022a). Nevertheless, the combination of the strong isotopic signal of irrigation water and the analysis that relies on relative isotopic changes in a brief period should limit the effect of the described biases on the results. The isotopic signals of stem samples collected 1 d after the irrigation had already reverted to their pre-experiment values, suggesting that the initial influx of labelled water was quickly removed from the trees despite the low evaporation demand.

The $\delta^2$H peak in spruce bulk stem water that was detected after 15 d was presumably linked to transpiration, as comparable tree water transit times, linked to a slow tree water transport velocity, have been observed in different settings: after 12 d in Magh et al. (2020), 14 d in Seeger and Weiler (2021) and 9–18 d in Nehemy et al. (2022b). Strong depletion of the bulk stem water isotopic signals after the snowmelt event, comparable to both the bulk soil and the lysimeter water isotopic signals present at the time, suggested that the homogenizing effect of snowmelt is not limited to the soil but extends throughout the soil–vegetation continuum. Although an isotopic offset between soil waters and bulk stem water is typically observed in cool wet environments (de la Casa et al., 2022), this may not be the case in the brief period following the snowmelt. Collected data show that full or partial refilling of the tree water storage can typically occur during large infiltration events, such as the snowmelt, and affect the seasonal isotopic cycle of sub-arctic forests. Knighton et al. (2020) found that simulating tree storage, especially during the periods of low transpiration, can be relevant for the estimation of tree water dynamics and water residence times in critical zone.

## 5 Conclusions

Our "double-irrigation" experiment, where the experimental plot was first saturated with an isotopically highly enriched signal and afterwards flushed by the isotopically depleted snowmelt signal, allowed us to infer infiltration and soil water mixing mechanisms in shallow sub-arctic till soil. We propose a conceptual model with three infiltration stages that are related to the amount of infiltrating water and the heterogeneous subsurface flow and eventually lead to complete soil water displacement. Effectively, the model considers that a major fraction of the infiltrating waters largely bypasses the soil matrix and moves via a macropore network during large infiltration events, and the connection between different soil compartments is re-established through groundwater rise. In the first stage, the surface microtopography prompts focused infiltration, where soil water flows vertically through an unsaturated macropore network, bypassing the soil matrix and recharging the groundwater storage. The second stage is defined by groundwater rise through the macropore network, gradual filling of the macropores and increased horizontal hydrological connectivity in the soil, followed by the establishment of shallow subsurface lateral fluxes. During the third stage, soil matrix water replenishment occurs under long-lasting soil saturation and high groundwater table conditions, which promote matrix flow and pore water exchange. In sub-arctic areas, the homogenization effect can also extend to the stem water.

A massive but short-lasting infiltration event, accompanied by a quick groundwater table rise, produced highly heterogeneous conditions in the soil and negligible effects on the isotopic composition of the soil matrix below 45 cm depth. However, the snowmelt event was characterized by the progressive flushing of more mobile soil water and subsequent homogenization of the soil matrix. We found that the soil saturation caused water inflow into the root systems of nearby trees despite the very low radiative forcing. Our results highlight the unique role of snowmelt in soil water replenishment and isotope dynamics at the soil–vegetation interface. Our findings stress the importance of improving the understanding of present soil water recharge processes. Quantifying the soil water dynamics and sources is critical for quantifying future infiltration patterns in high-latitude environments that are particularly sensitive to climate change.

*Data availability.* Soil water and groundwater isotopic data are available at https://etsin.fairdata.fi/dataset/964149fe-7718-4929-95e9-9deb406d96de (Ala-Aho et al., 2023).

*Supplement.* The supplement related to this article is available online at: https://doi.org/10.5194/hess-28-1-2024-supplement.

*Author contributions.* BK, HM and PA-A were responsible for funding acquisition, study conceptualization and project administration. PA-A, HM, FM and MS conducted fieldwork, FM was responsible for lab work, and all the authors were involved in data analysis. FM was responsible for data visualization and the preparation of the original draft. PA-A, HM and MS reviewed and edited the draft.

*Competing interests.* The contact author has declared that none of the authors has any competing interests.

ther geographical representation in this paper. While Copernicus Publications makes every effort to include appropriate place names, the final responsibility lies with the authors.

*Acknowledgements.* The study was funded by the Kvantum Institute at the University of Oulu and supported by a Maa- ja Vesitekniikan Tuki ry grant (43562) and the KH Renlund Foundation. Pertti Ala-Aho was supported by the Academy of Finland grants 316349 and 347348. Hannu Marttila was supported by Academy of Finland grants (318930, 316014, 347704, 346163, 337523 and 316349), Profi 4 Arcl and the National Freshwater Competence Centre (FWCC). Matthias Sprenger thanks the German Hydrological Society (DHG) for the travel support through their DHG Feldstipendium. The authors thank Valtteri Hyöky, Kashif Noor, Stephanie Gerin, Juho Kinnunen, Danny Croghan and Päivi Pietikäinen for their help with sample collection and Annalea Lohila for her help with the instrumentalization of the experimental plot. The authors thank Masaki Hayashi and an anonymous reviewer, whose comments significantly improved the quality of the paper.

*Financial support.* This research has been supported by the Research Council of Finland (grant nos. 316349, 318930, 316014, 347704, 346163 and 337523) and the Maa- ja Vesitekniikan Tuki Ry (grant no. 43562).

*Review statement.* This paper was edited by Markus Weiler and reviewed by Masaki Hayashi and one anonymous referee.

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
