# Peer review of "Snowmelt-mediated isotopic homogenization of shallow till soil"

_EGUsphere, 2023_

## Author Comment (AC1)

RC1:

**Summary:** A "double" irrigation experiment was undertaken where first an experimental plot was irrigated with waters enriched in deuterium, followed by isotopically depleted snowmelt infiltration under natural conditions. Through soil water, groundwater, and soil sampling, a conceptual model of macropore and matrix storage/flow was developed which brings new insights into the physical processes of infiltration and the role of snowmelt on vadose zone hydrology. The quality of the writing and extensive literature review incorporated throughout the manuscript is excellent. I have just a few general and specific comments for consideration.

We appreciate your positive assessment of our work, and suggestions for improving the manuscript quality. Please find our point-to-point response to the comments, questions and suggestions below.

**General:**

**1.** It's unclear how the tree response data is supportive of the overall theme of the paper, which largely focuses on macropore infiltration processes and soil storage dynamics. Presently it feels a bit out of place. It may be worth asking, "What does this information add to the story?" If this isn't clear, consider removing. While I recognize the importance of plant uptake as a pathway for removing soil water, I think additional efforts should be made to tie this piece together with the other components of the manuscript if it is to be kept. This includes:

- Abstract
- The introduction: The manuscript is well positioned in the literature from a macro-pore/soil storage perspective, but considering vegetation dynamics are part of the research questions, this should be better introduced.
- Discussion: Greater discussion on the importance of the tree uptake in the homogenization effect or other new/critical insights gained
- Conclusions: There should be a conclusion for each research question/objective and this is not present in the conclusions

Thank you for the suggestion, we agree that the overall presentation of the stem water data should be improved and linked to the rest of the manuscript in a better way. Although the amount stem water data is somewhat limited, compared to the amount of soil water data in this study, we feel that these findings should be included as they are important in the context of cold climate and seasonally snow-covered areas. Including the stem water data also ensures a more holistic view of the whole soil-plant continuum, rather than just the soil, and further shows that the homogenization effect of snowmelt is not only limited to the soil but extends to the plant water. Also, the literature regarding the stem water response to massive infiltration events in northern regions is currently scarce, although such events are particularly important for stem rehydration and happen on a yearly basis in the form of snowmelt.

To properly address the stem water dynamics part of the experiment, more details will be added to the Introduction section to provide a clear idea why we consider the stem water dynamics important in the context of this manuscript, and a further commentary later in Section 4.4. A summary of the stem water dynamics will be added to both Abstract and Conclusion sections.

**2.** Although in this case, the soil temperatures remained above freezing, this will not be the case for many cold regions throughout the winter. How do you expect your conceptual model to change in years where seasonal frost develops at this site and for other sites with frozen soils?

Due to the location of the study, infiltration into the frozen soil was considered when creating the model, so we presume that the model is very well suited to the regions with seasonal soil freezing. The main mechanisms behind the water movement through the frozen soils, focused on infiltration (through the patches of unfrozen soil) and macropore flow, are also the main mechanisms used in the first stage of the proposed frost-free model. The difference is that most of the soil matrix in the upper soil layer is frozen, so the onset of matrix flow occurs faster.

Furthermore, the second infiltration stage in the model is linked to the overall reduction of the downward transport capacity of the soil, and a general reduction of the infiltration capacity is one of the main consequences of soil freezing. The freezing front that extends into the deeper soil in the form of "fingers" of variable lengths can potentially negatively affect the horizontal hydrologic connectivity in the affected soil zones, by creating small impermeable zones. Such an effect should largely be offset by the overall reduction in the size of the unfrozen zone, due to the freezing of the upper soil layers, which constrains the movement and lateral fluxes of soil water fluxes to a smaller zone, thus enabling an increased connectivity.

The third stage, soil matrix refilling, depends on the height of the groundwater table and can be limited by the depth of the freezing front. This effect should be negligible in the case of shallow soil freezing, as the isotopic composition of uppermost soil layers shifts towards the infiltration water signal almost instantaneously.

We will formulate the above as an additional discussion paragraph about the applicability of the model in frozen soil conditions.

**Line Specific:**

**Figure 1:** The subscripts in 1a should be enlarged for clarity. The co-ordinates around the border of 1b should be removed or enlarged so they can be read. They are presently illegible, even zoomed in.

Subscripts will be enlarged and highlighted, and the coordinates removed.

**Line 72:** Some more background on the potential responses previously reported might help set up this research question a little better. At present it reads a bit disconnected from the other questions and introduction.

More details were provided in response to General 1 comment.

**Line 116:** What area do the sprinklers cover and where were they positioned relative to your sample locations? locations could be added to Figure 1.

The location of sprinklers and the irrigated area will be highlighted in revised Figure 1.

**Line 126:** I think this information would be more helpful presented with the soil core sampling. I was left wondering when you sampled the cores after reading this section, but then realized these detailed were in 2.3.

We will explicitly mention when and how the soil cores from the EP were collected during the experiment, and make a clear distinction between coring done during the long-term monitoring and during the experiment.

**Line 143:** Since it is common to include Supplementary Data in a supporting document/appendix, I suggest renaming Section 2.4 to avoid confusion.

Section 2.4 name will be changed to "Additional data".

**Figure 2 and 3:** include all data in the legend, not just in the caption. It is difficult to interpret as is. Suggest switching the colours between 30 and 35 cm so there is a gradation of colour from light to dark = shallow to deep. Also need a legend for (2d).

The legends will be added, and the colors altered.

RC2:

GENERAL COMMENTS

The manuscript presents an interesting study using deuterated water to observe the response of a forested hillslope to an application of a large quantity of water mimicking a snowmelt event, and to understand the interaction between soil matrix and macropore network. The field experiment is designed well and the data are of high quality. The manuscript is well organized and written in most parts, and the methods are clearly described. The study has strong potential to make a unique and significant contribution to hydrological science. However, I see a substantial room for improvements in the analysis and discussion of the results regarding the soil water dynamics. The manuscript could use more careful and quantitative interpretation based on the principles of soil physics. As it is written, some of the data interpretation is speculative and may not support the main conclusions. I will elaborate more in my specific comments below.

Thank you for an encouraging assessment of our work. The detailed suggestions helped us provide more quantitative, rather than descriptive data analysis, although the final extent of the analysis is limited by the number of observations relative to the strong soil heterogeneity. Please find our point-to-point response to the comments, questions and suggestions below.

SPECIFIC COMMENTS

Line 27. Unsaturated (vadose). The unsaturated zone and the vadose zone are not interchangeable. Please select the term that is most suitable for the context of this study and use it consistently throughout.

"The vadose zone is frequently called the unsaturated zone" (Nimmo, 2009, Encyclopedia of Inland Waters (2009), vol. 1, pp. 766-777, https://wwwrcamnl.wr.usgs.gov/uzf/abs_pubs/papers/nimmo.09.vadosewater.eiw.pdf). For consistency, we will use "vadose zone" throughout the manuscript.

Figure 1. Please delineate the actual area of irrigation using a polygon or rectangle. Open white squares and light gray squares are shown in Fig. 1a but not explained. Please include them in the legends and explain them in the figure caption. Please indicate latitudes and longitudes in Fig. 1c. Please use appropriate font sizes in these figures keeping in mind that Figure 1 may be published in a reduced size.

As mentioned in the response to Reviewer 1, the formatting of Figure 1 will be improved.

Line 90-91. Please pay attention to the number of significant digits in comparison to the accuracy of measurements.

The number of significant digits will be reduced to 1.

Line 117. 0.24. This is reported as a unitless number, but it should be either density (kg/m3) or snow water equivalent (mm).

The unit will be changed to 240 mm.

Line 120. 163.6 mm. How was this determined? Was it measured using one precipitation gauge (Fig. 1a)? If so, how representative is it of the entire irrigated area? Unless the sprinkler system has a uniform rate of application over the entire area, one would normally need multiple precipitation gauges to estimate an average. Please explain the uncertainty in this value. Was there an attempt to calculate the application rate from the volume of water going through the sprinkler system?

We used the sprinkler setup that was previously installed by Korkiakoski et al., 2022. section 2.2 (doi: 10.5194/BG-19-2025-2022): They positioned sprinklers in the plot so that irrigated water can evenly reach their every measurement point, and their measurement points were generally more spatially distributed than the ones used in our experiment. They checked the spatial distribution of irrigation with rain gauges and plastic buckets, as the amount of water in each bucket was proportioned to the rain gauges (in mm) based on their dimensions. The irrigated area had 3–5.5 m width and 10–21 m length, depending on the wind conditions. This information will be added to the methods section.

The wind speed was relatively low (2.98 m/s on average, with occasionally wind gust up to maximum of 9.5 m/s) and wind direction stable (~200-250 degrees) during the irrigation experiment. Overall, 20 tanks of 1000 L volume were applied to the plot, and 163.6 mm recorded by the rain gauge on the far side of the plot proves that most of the water reached all the parts of the plot. A network of soil moisture sensors at 7 cm depth, installed within the experimental plot by FMI (see figures below), as well as visual observations of the surface water ponding (Fig. S4 in Supplement) in all parts of the plot, confirm this. While the spatial distribution of the irrigation was not perfect, the main goal of the experiment, i.e. simulating a large magnitude infiltration event that significantly alters the antecedent soil isotopic signal, was achieved.

[Figure]

Figure1. Sketch of the planned FMI sensor locations within the experimental plot. Note that most sensors were not installed. Profiles S2 and S3 have been added for the reference.

[Figure]

Figure 2. Volumetric soil moisture content in the experimental plot at 7 cm depth recorded by FMI sensors.

These figures will be reworked and included in the Supplement.

Line 123. 600 kPa. This is roughly 6 bar. What was the bubbling pressure of the suction cup? How was this value chosen? Does it represent the actual condition of soil pores? Please explain.

Thank you for noticing this. There is an editing lapse in manuscript at lines 123 and 137, the listed values should be in hPa, rather than kPa. The real value is 60 kPa, which is equal to the bubbling pressure of the suction cup (https://www.dmr.eu/product/prenart-super-quarts-suction-cells/). The value of the applied suction was chosen based on the values that have been commonly used in isotope studies, such as Brooks et al., 2010, (doi: 10.1038/ngeo722) for the purpose of comparability.

Line 156. Bulk soil water. Does this mean the bulk of an entire core? Or, does it refer to individual core sections? What does 'bulk' mean in the context of this study? Please explain.

We define "bulk water" in line with other isotope related soil water work (e.g, Geris et al., 2017, doi: 10.1016/j.scitotenv.2017.03.275, Sprenger et al. 2018, doi: 10.2136/VZJ2017.08.0149) as the water that represents a mix of all water stored in the soil, from matrix water to highly mobile water. The real

amount of extracted water can vary depending on the clay content (negligible in our case) and soil matric potential. With the currently available soil water isotope sampling methods it is not possible to make a more clear/precise definition of the sampling range and associated water amount. An explicit definition of "bulk soil water" will be added to the manuscript. We always refer to the individual 5 or 10 cm core sections, and this information will also be added to the manuscript.

Figure 2c. This figure is difficult to comprehend. Symbols and colours are difficult to differentiate, and there are too many data series in one graph. Please improve the presentation. At least, more clearly distinguishable symbols and colours need to be used. Pale yellow is not easy to see on the white background. Also, it will be easier for the reader if Profile 2 and Profile 3 are presented in separate graphs.

The formatting will be adjusted.

Figure 2d. This figure is also difficult to comprehend. Please improve the presentation.

As mentioned previously, the color scheme will be changed.

Line 186. This sentence describes the response of 35-cm depth. However, I see that the 60-cm sensor responded before the water table started to rise, but this sampler was located far above the water table. This seems contrary to the sentence. Please explain. Overall, this paragraph could use a clearer writing that is consisted with the data presented in figures.

The fact that the 60-cm sensor had the fastest response is mentioned at lines 196-197. The idea behind mentioning the pan lysimeters at line 186 was to show that they only responded to the irrigation when the groundwater level was above their installation depth. This was further elaborated in section 4.2, at lines 372-377. We understand that the current text arrangement can cause confusion, so the sentence from line 186 will be moved to the end of section 3.1.

Line 190. Orange and yellow are difficult to distinguish. Please use different colour schemes.

The color scheme will be changed.

Line 192. Brown vertical lines. These lines look more purple than brown. Please include this in the graph legends, instead of describing it verbally.

The legend will be reworked.

Line 202. What was the date of another irrigation study? Was it before or after the study in the manuscript?

It happened before this study. From Korkiakoski et al., 2022: *The irrigation periods were 28 May to 7 September 2018 and 6 June to 29 August 2019*. Note that these dates also include this experiment with deuterated water, since Korkiakoski et al., have taken samples after our irrigation. Their non-labelled irrigation was stopped a few weeks before our experiment started. We will clarify this in the methods section of the revised manuscript.

Line 203. What was the deuterium value of the tap water? Please indicate it here.

Based on the 46 tap water samples from 2018, the average d$^2$H value was -104.36 ‰, with a standard deviation of 0.45 ‰. This information will be added to the methods section of the revised manuscript.

Line 220. Soil moisture levels. Pale yellow lines are difficult to see on white background. Please use a different colour.

As mentioned previously, the color scheme will be changed.

Line 238. Reference plot. This is described as 'natural soil' in the legend of Fig. 5d. Please use a consistent term.

The legend of Fig. 5d. will be adjusted accordingly.

Line 284. Infiltration capacity of surficial soil matrix is exceeded. This is an example of statement that is lacking quantitative consideration. Infiltration capacity of surface soil is quite high when it is unsaturated and decreases as the soil becomes saturated. The lower limit of infiltration capacity is the saturated hydraulic conductivity, which is on the order of 10^-6 to 10^-5 m/s (Line 90-91). These were determined using a falling-head permeameter on soil samples, implying the soil matrix conductivity. The intensity of irrigation was generally less than 10 mm/h or 3 x 10^-7 m/s (Figure 2a). Given this, it is not clear how the infiltration capacity of the surficial soil matrix can be exceeded. Please present a convincing explanation.

We used metal cylinders to collect undisturbed soil samples for the falling-head permeability test, so they can contain macropores too, which should result in higher conductivity than the soil matrix itself. But we do agree that infiltration capacity might not be the only relevant factor in this case. Another limiting factor is the water holding capacity of the surface soil layers, which is relatively low due to the high porosity and low bulk density. A figure with pF curves is attached for the reference. It is clear that the field capacity of the surface soil can be exceeded, even for infiltration events of lower intensity than 10 mm/h.

We will add a mention of this to the manuscript discussion part and highlight the role of the low water holding capacity of surface soil layers. A Figure with pF curves will be included in the Supplement.

[Figure]

Figure 3. pF curves

Line 285-286. Macropore flow is unsaturated. What is the evidence of unsaturated condition? The soil moisture sensors measure soil matrix, and are insensitive to macropores that occupy relatively small volume. Again, please present a convincing explanation.

Direct observations of the macropore saturation were not made, due to the scale of the experiment. Unsaturated flow in macropores is assumed as the default condition, as observed in many previous studies. This should especially be the case in unsaturated conditions, as macropores tend to be hydrophobic relative to the soil matrix. During the first ~12 hours of the experiment, the soil moisture values were generally below their maximum and there was no surface water ponding, all together indicating that soil matrix was unsaturated, and majority of infiltrating water could propagate through the soil matrix. Nevertheless, bulk soil water isotopic signal, at least in surface soil layers, was continuously more enriched than the suction lysimeter signal (Fig. 3 and lines 209-2011), indicating that some amount of the enriched irrigation water was moving through larger pores.

Line 288. Groundwater rises toward towards the surface. The rise of the water table is indeed shown in Fig. 2b. However, Fig. 2d shows constant moisture content at 60 cm in Profile 2, which is located adjacent to the water-table monitoring well. This implies that the soil at 60cm was saturated before the water table rose to 60cm. This does not seem likely because the capillary fringe cannot be as thick as 50cm in near-surface soil. I feel that something is missing here. Please present more careful interpretation of soil water dynamics based on the principles of soil physics.

The soil moisture sensor at 60 cm depth in profile 2 generally remains stable during the whole observation period, even during the snowmelt event. Unfortunately, this points to the possibility of sensor malfunction. The measurements from this sensor will be entirely removed from the manuscript, as there is a reasonable doubt that it wasn't functioning properly. Furthermore, no conclusions were

previously made based on the measurements from this sensor, so its removal will not affect the discussion section. The note that these observations were removed will be added to the methods section.

Line 289. Groundwater exfiltration. The term exfiltration specifically means that groundwater is discharging at the ground surface. It seems unlikely that exfiltration was happening while the water table was below the ground surface (Fig. 2b). Was exfiltration visually observed? Please present an explanation.

Thank you for the suggestion. We mention that it is exfiltration to the upper soil layers, rather than the surface, but agree that changing the term to "advances" would make the terminology more accurate.

Line 308. Dashed red lines. The red lines indicate 30cm, not 5cm, in this figure. Please be consistent between the texts and figures.

The sentence will be restructured for more clarity. We refer to soil patches at both 5 and 30 cm depth, as it can be seen from Fig. 2d that soil moisture at both locations (solid yellow and dashed red) changes very little, despite the changes at other depths in the same profiles.

Line 311. Please refer the reader to Fig. 3.

A reference will be added.

Line 313. (Smaller pores in the soil matrix) is filled first. Figure 2d indicates only 5% increase in water content at 5-cm depth. This seems inconsistent with a major shift in isotopic composition depicted in Fig. 3. Does it make sense in terms of mass balance consideration? Please explain.

Referring to the period between -4 hours and +5 hours, at 0-10 cm soil depth. The initial conditions were ~35% volumetric soil moisture content with an average isotopic signal of ~ -77 ‰, which changed to ~40% and ~ +11 ‰, respectively. The average isotopic signal of the infiltrating water was ~ +72 ‰. Considering bulk density of ~1 g/cm$^3$ and simple two end-member mixing, we can estimate that a 5 % increase in soil moisture would change the overall isotopic composition to ~ -58 ‰. Nevertheless, the amount of water applied to the soil, 36.2 mm, was much higher than the amount needed to increase moisture by 5 %. It can be expected that a certain fraction of antecedent soil water was displaced by the infiltrating water, thus changing the ratio of enriched and depleted water. This ratio, and the final bulk soil isotopic composition, are also affected by the soil heterogeneity. Lastly, bulk samples of soil sections with high moisture content, taken during the irrigation event, also contain a fraction of more mobile irrigation water. This irrigation water produces a stronger isotopic than hydrometric (i.e., soil moisture response) effect. A mention of soil water displacement effect, and a brief discussion regarding the relationship between bulk soil water isotopic signal and soil moisture content will be included.

Line 323. Does lysimeters preferentially sample macropores? They are subjected to high magnitude of matric potential (600 kPa). Please interpret the data more carefully considering the actual function of lysimeters.

Suction lysimeters (at 600 hPa) should in theory sample soil water of lower mobility than the water in macropores, but of a higher mobility than the matrix soil water. The difference between suction lysimeter water and bulk and macropore soil water also depends on the soil moisture. According to this comment and the comment for line 156, regarding the bulk soil water definition, a short explanation related to different soil water sampling methods, and soil waters of different mobility that they sample, will be added.

Line 345. Inability to infiltrate deeper. Many of the papers cited in this paragraph were on frozen soil infiltration, but this experiment was conducted under unfrozen condition. What prevents infiltration when the irrigation rate was much below the saturated hydraulic conductivity of surface soil? Please present a convincing explanation.

Surface water ponding was observed after 12 hours, after over 60 mm of irrigation water was sprinkled onto the soil. Although the infiltration capacity of surface soil was maybe not exceeded, the infiltration capacity of heavily compacted, lower soil layers surely was exceeded. These low permeability layers, combined with strong spatial heterogeneity of the soil and low water holding capacity of the surface layer, result in surface water ponding. We will add this to the discussion section.

Line 354-355. This sentence compares the rate of water-table rise and the soil saturated hydraulic conductivity. I cannot understand the logical connection between the two quantities. The rate of water-table rise does not indicate groundwater flux. The water table can rise quickly with a small addition of water if the soil is nearly saturated in the capillary fringe. The rise in the water table can occur when the flow direction is downward. Please revise the conceptual model of the water table dynamics and reinterpret the data based on the principles of soil physics.

We acknowledge that groundwater rise coincides with an increase of the soil moisture at certain depths and profiles, and that fast groundwater level rise occurs during the soil wetting stage, thus showing a link between soil saturation in the capillary fringe and the rate of groundwater level increase. On the other hand, a strong correlation between the groundwater isotopic signal and the total amount of applied irrigation water (line 185) shows that irrigation water reaches groundwater storage without much mixing with the soil matrix water that still largely has more depleted isotopic signal. This indicates that the irrigation water mostly reaches and mixes with groundwater via macropores. The saturation of the capillary fringe should also result in the increase of the macropore flow (referring to line 346 in the manuscript and Jarvis, 2007, doi: 10.1111/j.1365-2389.2007.00915.x). 1-2 sentences about the groundwater level rise during the soil wetting stage, and the source of groundwater replenishment, will be added to the discussion section. The effect of soil saturation in the capillary fringe on groundwater dynamics will be included in the conceptual model.

Line 364. Upward water flow. This requires the matric potential gradient in excess of 1. Is there direct evidence of such a gradient? Please present the data.

There were no direct measurements of the matric potential, but here we refer to the groundwater level rise that mostly occurs via the macropore network. Macropores allow non-equilibrium flow that should be less influenced by the matric potential. The strongest evidence is derived from the pan lysimeter

dynamics, as there was no water in the pan lysimeters until the ground water level exceeded their installation depth (35 cm). Afterwards, large amounts of water were extracted from the collection bottles every hour. We do concur that a change of terminology is needed. "Upwards water flow" will be rephrased to "groundwater rise through the macropore network". Additionally, the mention of flow reversal at line 288 will be removed.

Line 378. Stumpp and Hendry (2012). This study was not conducted in a sub-arctic catchment. Please read the paper carefully and revise the sentence.

Thank you for the correction. The term sub-arctic will be replaced with seasonally snow-covered.

Line 397. Rothfuss et al. (2015). What were the findings, and where was the study conducted? Please add the information, so the reader can understand the context.

Rothfuss et al. (2015) found that the isotopic composition of the surface soil immediately shifts towards the isotopic composition of the infiltrating water. Clarification will be added to the text.

Line 407. Michelon et al. (2023). Where was this study conducted?

The study was conducted in a steep mountain catchment in Switzerland. Indeed, this is somewhat in contrast with the findings of Mueller et al., (2014), mentioned on line 418, and will be further disseminated in the discussion.

Line 431. Net radiation. Please indicate the unit in the graph.

The axis label will be changed accordingly.

---

## Author Response (AR1)

RC1:

**Summary:** A "double" irrigation experiment was undertaken where first an experimental plot was irrigated with waters enriched in deuterium, followed by isotopically depleted snowmelt infiltration under natural conditions. Through soil water, groundwater, and soil sampling, a conceptual model of macropore and matrix storage/flow was developed which brings new insights into the physical processes of infiltration and the role of snowmelt on vadose zone hydrology. The quality of the writing and extensive literature review incorporated throughout the manuscript is excellent. I have just a few general and specific comments for consideration.

We appreciate your positive assessment of our work, and suggestions for improving the manuscript quality. Please find our point-to-point response to the comments, questions and suggestions below. Text added to the manuscript is marked in red italics, while the text that remained the from previous version is marked with black italics. Line numbers correspond to the version with tracked changes.

**General:**

**1.** It's unclear how the tree response data is supportive of the overall theme of the paper, which largely focuses on macropore infiltration processes and soil storage dynamics. Presently it feels a bit out of place. It may be worth asking, "What does this information add to the story?" If this isn't clear, consider removing. While I recognize the importance of plant uptake as a pathway for removing soil water, I think additional efforts should be made to tie this piece together with the other components of the manuscript if it is to be kept. This includes:

- Abstract
- The introduction: The manuscript is well positioned in the literature from a macro-pore/soil storage perspective, but considering vegetation dynamics are part of the research questions, this should be better introduced.
- Discussion: Greater discussion on the importance of the tree uptake in the homogenization effect or other new/critical insights gained
- Conclusions: There should be a conclusion for each research question/objective and this is not present in the conclusions

To properly address the stem water dynamics part of the experiment, as suggested, more details were added to different sections of the manuscript.:

Abstract: *Extensive soil saturation induced the flow of labelled water into the roots of nearby trees.*

Introduction (lines 40-44): *Plant water uptake, as one of the most important factors of ecosystem functioning, can be dependent on both soil water compartmentalization (Brooks et al., 2010) and seasonal availability (Allen et al., 2019). On the other hand, soil water dynamics is also influenced by the vegetation, as soil water content and isotopic variability can be dependent on the plant cover (Oerter and Bowen, 2019), meaning that both soil and stem water observations are required to understand the fate of the water that infiltrates the soil.*

and (lines 63-64)

*The effect of high water availability during the period of low radiation forcing on seasonal variability of stem water in sub-arctic forests, which often occurs during the snowmelt season, is still underexplored.*

Discussion (section 4, lines 335-336): *Furthermore, the homogenization was not only limited to the soil, but it extended throughout the soil-vegetation interface.*

Discussion (section 4.4, lines 555-557): *We speculate that timing of the experiment allowed us to first detect quick and short-lasting stem water enrichment caused by fast soil saturation, and afterwards a more gradual enrichment that resulted from transpiration-related root water.*

and (lines 572-574)

*The isotopic signals of stem samples collected one day after the irrigation were already reverted to their pre-experiment values, suggesting that the initial influx of labelled water got quickly removed from the trees despite the low evaporation demand.*

and (lines 580-585)

*Although isotopic offset between soil waters and bulk stem water is typically observed in cool wet environments (de la Casa et al., 2022), this may not be the case in brief period following the snowmelt. Collected data show that full or partial refilling of the tree water storage can typically occur during large infiltration events, such as snowmelt, and affect the seasonal isotopic cycle of sub-arctic forests. Knighton et al. (2020) found that simulating tree storage, especially during the periods of low transpiration, can be relevant for estimation of tree water dynamics and water residence times in critical zone.*

Conclusion:

*In sub-arctic areas, the homogenization effect can also extend to the stem water.*

and

*We found that the soil saturation caused water inflow into the root system of nearby trees, despite the very low radiative forcing.*

**2.** Although in this case, the soil temperatures remained above freezing, this will not be the case for many cold regions throughout the winter. How do you expect your conceptual model to change in years where seasonal frost develops at this site and for other sites with frozen soils?

An additional discussion paragraph about the applicability of the model in frozen soil conditions was added to the Discussion (section 4.3, lines 535-547).

*To understand the limits of the conceptual model applicability, it is important to assess how the described mechanisms would work in the frozen soil conditions that are expected in the sub-arctic catchments. Although soil freezing in the EP was not directly observed during the experimental period,*

*we suggest that the proposed model is theoretically well suited to the frozen soil conditions. Namely, the main the mechanisms of water movement in frozen soil conditions, focused infiltration and macropores flow, are also key mechanism that were observed during the first infiltration stage in this study. The freezing of the upper soil layers is advantageous for macropore flow development and can possibly trigger the onset of macropore flow even faster than in the case of unfrozen soil. One of the requirements for the groundwater rise through the macropores, that defines the second infiltration stage, is that infiltration exceeds the transport capacity of the soil. As soil pores are getting filled by the ice, their overall volume and consequently soil transport capacity get reduced, again supporting the proposed infiltration mechanism. At the same time, the ice-filled pores can act as flow restricting zones that limit the hydrologic connectivity between different soil patches, especially in the surficial soil layer where more extensive freezing should be present. The depth of the freezing front could additionally limit the homogenization effect that is caused by shallow and long-lasting groundwater table, but this effect should however be limited to the surficial soil.*

**Line Specific:**

**Figure 1:** The subscripts in 1a should be enlarged for clarity. The co-ordinates around the border of 1b should be removed or enlarged so they can be read. They are presently illegible, even zoomed in.

Subscripts were enlarged and highlighted, and the coordinates removed.

**Line 72:** Some more background on the potential responses previously reported might help set up this research question a little better. At present it reads a bit disconnected from the other questions and introduction.

More context about the importance of stem water was added to the Introduction, as described in the response to the General 1 comment.

**Line 116:** What area do the sprinklers cover and where were they positioned relative to your sample locations? locations could be added to Figure 1.

The location of sprinklers and the irrigated area are highlighted in the revised Figure 1.

**Line 126:** I think this information would be more helpful presented with the soil core sampling. I was left wondering when you sampled the cores after reading this section, but then realized these detailed were in 2.3.

Sections 2.2 and 2.3 were reorganized, the experiment was described in section 2.2 and all sampling is described in section 2.3 (Data collection). The part about soil coring (lines 160-166):

*Soil cores from the EP were collected 4 times during the experiment, using a percussion drill (Cobra 148, Atlas Copco) with a window sampling tube extension (RKS with a reinforced cutting edge with a core cutter Ø80 mm x 1 m, GEOLAB Paweł Szkurłat). Two replicate cores were collected each time, and all cores were sampled at 5 cm increments down to a 50 cm soil depth and at 10 cm increments from a 50–100 cm depth. Furthermore, soil coring campaigns monitoring the seasonal changes in the EP were conducted 2 weeks after the experiment, at the peak snowpack in April 2020 and after 2020*

*snowmelt in mid-June. Soil cores immediately downslope of the EP were collected 1 day after the experiment, and again 2 weeks later and under deep snowpack in April 2020.*

**Line 143:** Since it is common to include Supplementary Data in a supporting document/appendix, I suggest renaming Section 2.4 to avoid confusion.

Section 2.4 name is changed to "Additional data".

**Figure 2 and 3:** include all data in the legend, not just in the caption. It is difficult to interpret as is. Suggest switching the colours between 30 and 35 cm so there is a gradation of colour from light to dark = shallow to deep.  Also need a legend for (2d).

The legends are added, and the colors altered.

RC2:

GENERAL COMMENTS

The manuscript presents an interesting study using deuterated water to observe the response of a forested hillslope to an application of a large quantity of water mimicking a snowmelt event, and to understand the interaction between soil matrix and macropore network. The field experiment is designed well and the data are of high quality. The manuscript is well organized and written in most parts, and the methods are clearly described. The study has strong potential to make a unique and significant contribution to hydrological science. However, I see a substantial room for improvements in the analysis and discussion of the results regarding the soil water dynamics. The manuscript could use more careful and quantitative interpretation based on the principles of soil physics. As it is written, some of the data interpretation is speculative and may not support the main conclusions. I will elaborate more in my specific comments below.

Thank you for an encouraging assessment of our work. The detailed suggestions helped us provide more quantitative, rather than descriptive data analysis, although the final extent of the analysis is limited by the number of observations relative to the strong soil heterogeneity. Please find our point-to-point response to the comments, questions and suggestions below. Text added to the manuscript is marked in red italics, while the text that remained the from previous version is marked with black italics. Line numbers correspond to the version with tracked changes.

SPECIFIC COMMENTS

Line 27. Unsaturated (vadose). The unsaturated zone and the vadose zone are not interchangeable. Please select the term that is most suitable for the context of this study and use it consistently throughout.

It was changed to vadose zone.

Figure 1. Please delineate the actual area of irrigation using a polygon or rectangle. Open white squares and light gray squares are shown in Fig. 1a but not explained. Please include them in the legends and explain them in the figure caption. Please indicate latitudes and longitudes in Fig. 1c. Please use appropriate font sizes in these figures keeping in mind that Figure 1 may be published in a reduced size.

As mentioned in the response to Reviewer 1, the formatting of Figure 1 is improved.

Line 90-91. Please pay attention to the number of significant digits in comparison to the accuracy of measurements.

The number of significant digits is reduced to 1.

Line 117. 0.24. This is reported as a unitless number, but it should be either density (kg/m3) or snow water equivalent (mm).

The unit is changed to 240 mm.

Line 120. 163.6 mm. How was this determined? Was it measured using one precipitation gauge (Fig. 1a)? If so, how representative is it of the entire irrigated area? Unless the sprinkler system has a uniform rate of application over the entire area, one would normally need multiple precipitation gauges to estimate an average. Please explain the uncertainty in this value. Was there an attempt to calculate the application rate from the volume of water going through the sprinkler system?

The information about the sprinkler system was added to the methods section 2.2 (lines 124-128):

*The sprinkler setup was installed and maintained by Korkiakoski et al. (2022), and sprinklers were positioned so that irrigation water can be distributed evenly within the EP, covering the area of 3-3.5 m width and 10-21 m length depending on the wind conditions. Weather conditions during the experiment were favourable, with relatively low wind speed (2.98 m/s on average, with occasional wind gust of up to 9.6 m/s), stable wind direction (mostly between 200 and 250 degrees) and no rainfall.*

and (lines 136-138)

*Although spatiotemporal distribution of irrigation water was not completely uniform, plot-wide surface water ponding and soil moisture increase showed that the main goal of the experiment, i.e. simulation of a high magnitude infiltration event, was achieved.*

Line 123. 600 kPa. This is roughly 6 bar. What was the bubbling pressure of the suction cup? How was this value chosen? Does it represent the actual condition of soil pores? Please explain.

The value was changed to 600 hPa.

Line 156. Bulk soil water. Does this mean the bulk of an entire core? Or, does it refer to individual core sections? What does 'bulk' mean in the context of this study? Please explain.

A paragraph describing assumptions for different water samples was added to the section 2.3 Data collection (lines 170-177):

*We define the bulk soil water in line with other isotope related soil water works such as Geris et al. (2017), as the water extracted from the soil that represents a mix of all waters stored in the soil, from soil matrix water to highly mobile water. Suction lysimeters are assumed to sample the soil waters of lower mobility than the water moving through macropores, but higher mobility than the soil matrix water. The difference in isotopic signal between bulk soil samples and suction lysimeter samples represents a combination of tightly bound soil matrix water and macropore water isotopic signals, depending on the soil moisture content. The isotopic signal of pan lysimeter water is assumed to be the most realistic representation of highly mobile soil water. Bulk stem water is considered to reflect a mixture of various stem water pools that contain waters of different ages.*

Figure 2c. This figure is difficult to comprehend. Symbols and colours are difficult to differentiate, and there are too many data series in one graph. Please improve the presentation. At least, more clearly distinguishable symbols and colours need to be used. Pale yellow is not easy to see on the white background. Also, it will be easier for the reader if Profile 2 and Profile 3 are presented in separate graphs.

The formatting is adjusted.

Figure 2d. This figure is also difficult to comprehend. Please improve the presentation.

As mentioned previously, the color scheme is changed.

Line 186. This sentence describes the response of 35-cm depth. However, I see that the 60-cm sensor responded before the water table started to rise, but this sampler was located far above the water table. This seems contrary to the sentence. Please explain. Overall, this paragraph could use a clearer writing that is consisted with the data presented in figures.

The sentence is moved to the end of section 3.1 (lines 258-260).

Line 190. Orange and yellow are difficult to distinguish. Please use different colour schemes.

The color scheme is changed.

Line 192. Brown vertical lines. These lines look more purple than brown. Please include this in the graph legends, instead of describing it verbally.

The legend is reworked.

Line 202. What was the date of another irrigation study? Was it before or after the study in the manuscript?

The information is now included into the manuscript, section 3.1 (lines 246-248):

*The EP was used in another irrigation study in 2018 and 2019, with an aim of to increase the overall soil water content at the plot (Korkiakoski et al., 2022). Due to the and thus frequently irrigated using tap water with a constant isotopic signal. As the study ended only several weeks before the onset of this study, bulk soil water d$^2$H depth profile variability (Fig. 3) was relatively low at the start of the experiment.*

Line 203. What was the deuterium value of the tap water? Please indicate it here.

Based on the 46 tap water samples from 2018, the average d$^2$H value was -104.36 ‰. The information is added to section 2.2 (Irrigation experiment, line 123) of the revised manuscript.

Line 220. Soil moisture levels. Pale yellow lines are difficult to see on white background. Please use a different colour.

The color scheme is changed.

Line 238. Reference plot. This is described as 'natural soil' in the legend of Fig. 5d. Please use a consistent term.

The legend of Fig. 5d. is adjusted.

Line 284. Infiltration capacity of surficial soil matrix is exceeded. This is an example of statement that is lacking quantitative consideration. Infiltration capacity of surface soil is quite high when it is unsaturated and decreases as the soil becomes saturated. The lower limit of infiltration capacity is the saturated hydraulic conductivity, which is on the order of 10^-6 to 10^-5 m/s (Line 90-91). These were determined using a falling-head permeameter on soil samples, implying the soil matrix conductivity. The intensity of irrigation was generally less than 10 mm/h or 3 x 10^-7 m/s (Figure 2a). Given this, it is not clear how the infiltration capacity of the surficial soil matrix can be exceeded. Please present a convincing explanation.

The first stage description is reworded (Section 4, lines 344-347):

*1)      First stage – the initiation of macropore flow: Surface microtopography induces surface ponding and focused infiltration. Infiltrating waters first fill the surficial soil matrix, but due to the*

*limitations in infiltration and water holding capacity of the surficial soil matrix, the water fluxes are conveyed downwards through the macropore network. The fluxes are largely vertical and macropore flow is unsaturated. Soil matrix in deeper soil layers is bypassed.*

A Figure with pF curves is included in the Supplement as Fig. S6.

Additional comments regarding the infiltration capacity are included in section 4.1 (lines 400-401):

*The transfer of flow towards the macropore network is further increased by the low water holding capacity of surficial soil at Pallas (Fig. S6).*

and (lines 415-416)

*These factors of heterogeneity explain how the observed surface water ponding could occur in isolated patches of soil although the irrigation rate was lower than the measured infiltration capacity of surficial soils.*

and (lines 421-425)

*As the irrigation progressed, and over 60 mm of water was applied to the EP, the movement of water through the soil was no longer only limited by the infiltration capacity of the surficial soil, but by the infiltration capacity of deeper soil layers too. As previously infiltrated water reached deeper, strongly compacted soil layers, its further percolation was halted and subsequent infiltrating water started accumulating in the upper soil layers, eventually exfiltrating to the surface.*

Line 285-286. Macropore flow is unsaturated. What is the evidence of unsaturated condition? The soil moisture sensors measure soil matrix, and are insensitive to macropores that occupy relatively small volume. Again, please present a convincing explanation.

Short discussion about the macropore saturation was added to section 4.2 (lines 463-473):

*Although direct observations of macropore flow were not possible due to the scale and setup of the study, the enrichment of bulk soil water isotopic signal in the upper layers, compared to the isotopic signal of water from suction lysimeters, can serve as an indicator that the water of higher mobility was present in the soil and moved through the macropore network already during the first hours of the irrigation. The isotopic enrichment observed at 60 cm depth (Fig. 2b) during the irrigation, that occurred before the enrichment in upper soil layers, further shows that preferential flow pathways were active even in the early stages of the experiment. On the other hand, the abrupt appearance and disappearance of large quantities of water in pan lysimeters, observed both during the irrigation experiment and the snowmelt, can be used to infer lateral flows and the near saturation of the macropore network.*

Line 288. Groundwater rises toward towards the surface. The rise of the water table is indeed shown in Fig. 2b. However, Fig. 2d shows constant moisture content at 60 cm in Profile 2, which is located adjacent to the water-table monitoring well. This implies that the soil at 60cm was saturated before the water table rose to 60cm. This does not seem likely because the capillary fringe cannot be as thick as

50cm in near-surface soil. I feel that something is missing here. Please present more careful interpretation of soil water dynamics based on the principles of soil physics.

The soil moisture sensor at 60 cm depth in profile 2 was removed from the graphs.

More discussion about groundwater rise was added to section 4.2 (443-460).

*A much faster groundwater response can occur in nearly saturated soils in the capillary fringe. As the groundwater level in the EP is typically at least 1 m below the ground during the summer and autumn (Fig. 4b), the capillary rise in sandy till soils in the EP should be limited to the soil layers at similar depth. Despite this, we observed that the groundwater level increased by 77 cm during an 8-hour period, between hour 6 and hour 14 of the experiment. The observed rate of groundwater rise and isotopic signal enrichment indicate that groundwater moves via bypass flow through the macropore network. In this way, a hydrological connection between infiltrating water and groundwater is established through the macropore network and a flow constricting zone in deeper soil layers is largely bypassed. A strong correlation between the total amount of irrigated water applied to the EP and groundwater isotopic signal shows that the enriched irrigation water reaches groundwater directly through the macropore network. The increase of soil moisture during the soil wetting stage is clearly an important driver and a necessary precondition of the groundwater level rise, based on the fact that they showed similar dynamics during the entire observation period (Fig. 4b and c). Still, generally high soil moisture and correspondingly low soil matric potential in the EP helped convey the infiltrating water towards macropores rather than in the soil matrix and limited the pore water exchange between soil matrix and macropores. Although macropore flow is typically unsaturated due to its high transport capacity (Cey and Rudolph, 2009; Jarvis, 2007) and relative hydrophobicity compared to the soil matrix, we can assume that the groundwater rise through the macropore network signals the temporary excess of water in the soil and near-saturation of vertical macropores, which consequently intensifies the lateral movement of water and strengthens horizontal hydrological connectivity in the macropore network.*

Also, the description of stage 2 in section 4 (lines 348-351) was modified:

*2) Second stage – development of lateral fluxes: As the transport capacity of certain parts of the macropore network in deeper soil is surpassed, and infiltrating water percolates towards the soil layers with lower permeability, groundwater rises towards the surface through the macropore network.*

Line 289. Groundwater exfiltration. The term exfiltration specifically means that groundwater is discharging at the ground surface. It seems unlikely that exfiltration was happening while the water table was below the ground surface (Fig. 2b). Was exfiltration visually observed? Please present an explanation.

The term is changed to "advances".

Line 308. Dashed red lines. The red lines indicate 30cm, not 5cm, in this figure. Please be consistent between the texts and figures.

The sentence was restructured (lines 374-379):

*The amount of water applied to the soil during this experiment was relatively large, considering the shallow soil at this location, yet soil water storage continuously displayed a strong heterogeneity in soil moisture and isotopic measurements in the two observed profiles. Our data shows that some soil patches at a 5 cm depth remained largely isolated from infiltrating waters for a long period (solid yellow lines in Fig. 2c and Fig. 2d), despite isotopic changes in deeper soil layers at the same profile already taking place (solid red lines in Fig. 2c and Fig. 2d) and plot-wide surface water ponding.*

Line 311. Please refer the reader to Fig. 3.

A reference is added.

Line 313. (Smaller pores in the soil matrix) is filled first. Figure 2d indicates only 5% increase in water content at 5-cm depth. This seems inconsistent with a major shift in isotopic composition depicted in Fig. 3. Does it make sense in terms of mass balance consideration? Please explain.

A short discussion, describing possible reasons for difference in hydrometric and isotopic response was added to section 4.1 (lines 384-392):

*During the first 5 hours of the experiment, the soil moisture content in first 10 cm of depth increased by less than 10 %, while the bulk soil $\delta^2H$ values increased by some 90 ‰ (from -77 to +11 ‰). Such $\delta^2H$ increase could not be caused by simple mixing of the antecedent soil water with the infiltrating water, as the amount of antecedent water (~35 %) was much higher than the soil moisture increase. This indicates that the process of soil water displacement in the upper soil layers was initiated in the early stages of the experiment, as the enriched water started entering the soil matrix and altering the isotopic signal of the soil water. It should be noted that the observed soil water enrichment could not only be caused by mixing and displacement processes, as the bulk soil water also contains a certain fraction of very mobile infiltrating water, which can further skew the isotopic values towards the enriched values.*

Line 323. Does lysimeters preferentially sample macropores? They are subjected to high magnitude of matric potential (600 kPa). Please interpret the data more carefully considering the actual function of lysimeters.

As described previously, a paragraph describing assumptions for different water samples was added to the section 2.3 Data collection, lines 170-178.

Line 345. Inability to infiltrate deeper. Many of the papers cited in this paragraph were on frozen soil infiltration, but this experiment was conducted under unfrozen condition. What prevents infiltration when the irrigation rate was much below the saturated hydraulic conductivity of surface soil? Please present a convincing explanation.

The infiltration capacity was addressed in our previous answer to the question regarding the line 284.

Line 354-355. This sentence compares the rate of water-table rise and the soil saturated hydraulic conductivity. I cannot understand the logical connection between the two quantities. The rate of water-table rise does not indicate groundwater flux. The water table can rise quickly with a small addition of

water if the soil is nearly saturated in the capillary fringe. The rise in the water table can occur when the flow direction is downward. Please revise the conceptual model of the water table dynamics and reinterpret the data based on the principles of soil physics.

The groundwater table rise was addressed in the answer to the question regarding the line 288.

Line 364. Upward water flow. This requires the matric potential gradient in excess of 1. Is there direct evidence of such a gradient? Please present the data.

"Upwards water flow" is rephrased to "groundwater rise through the macropore network". Additionally, the mention of flow reversal at line 288 (now line 350) is removed. Figure 7 was modified to incorporate the changes.

Line 378. Stumpp and Hendry (2012). This study was not conducted in a sub-arctic catchment. Please read the paper carefully and revise the sentence.

The term sub-arctic is replaced with seasonally snow-covered.

Line 397. Rothfuss et al. (2015). What were the findings, and where was the study conducted? Please add the information, so the reader can understand the context.

Clarification is added to the text (lines 498-499):

*The deuterated water signal was fully removed from the topsoil after rainfall events following irrigation (Fig. 5), consistent with the findings of Rothfuss et al. (2015) who identified that isotopic composition of surface soil immediately shifts towards the isotopic composition of the infiltrating water.*

Line 407. Michelon et al. (2023). Where was this study conducted?

The paragraph was restructured (lines 527-534):, and included in a wider discussion about the model applicability:

*Prolonged soil saturation is also less likely to occur on steep slopes, as Mueller et al. (2014) found highly spatially variable soil isotopic signals during the snowmelt period. However, they note that soil sampling campaign in their study took place roughly 4 months after the snowmelt, which is arguably longer time span than the transit times in upper 1 m of the soil. In a more recent study conducted in a steep alpine catchment, Michelon et al. (2023) recognized the potential of snowmelt to strongly flush the entire subsurface system and reset the isotopic values of soil waters. Furthermore, homogenization of soil waters during snowmelt should be expected in topographical lowlands with shallow groundwater table, where groundwater – surface water interactions are commonly observed in sub-arctic (Autio et al., 2023).*

Line 431. Net radiation. Please indicate the unit in the graph.

The axis label is adjusted.

---

## Author Response (AR2)

GENERAL COMMENTS

The authors addressed many of my comments on the original manuscript, which improved the clarity of texts and figures. However, a few comments have not been adequately addressed. I would recommend withholding the acceptance of the manuscript until the following issues are sufficiently addressed.

We thank the Reviewer for allocating the time for reviewing the manuscript and providing numerous comments which helped us improve the quality of the manuscript. We have now addressed all the issues raised by the reviewer. The line numbers regarding the changes made, reported in this response letter, correspond to the manuscript version with tracked changes (Muhic_et_al_2023_revision2_markups)

SPECIFIC COMMENTS

Figure 1. Please include legends for open white squares and light gray squares in Figure 1a.

Response to the Reviewer:

The light gray and white squares are covered and open metal collars for soil methane flux measurement, as photographed by the drone, that remained in the soil after the previous experiments which are not directly related to this study.

They are not included in the legend as they are not relevant to this experiment and their inclusion could be somewhat misleading. It is also not possible to remove them as they are a part of the original/raw photograph, rather than symbols (although they look "too regular/rectangular" in Figure 1 (a) of the manuscript which might be causing the confusion). Both open and closed collars can be seen on the left side photograph below that was taken during the post experiment bulk soil sampling, and open collar is shown or the right side photograph, taken during the experiment.

[Figure]

Changes made to the manuscript text:

Clarification was added to the Figure 1 caption at lines 88 – 91: "*Aerial photograph by Bastian Steinhoff–Knopp (Leibniz University Hannover, September 2018). The grey and white rectangles seen along the boardwalk are soil methane flux measurement collars that were installed prior to our experiment, remained at the study site, but did not have a function during the experiment*"

Line 133. This was Line 120 in the original manuscript concerning the total amount of irrigation (163.6 mm) recorded by the tipping bucket rain gauge. According to the authors' explanation, the irrigated area is 3-3.5 m by 10-21 m, which translates to 30-73.5 m2. On this area, 20 times 1000 L (= 20 m3) of water was applied, meaning that the possible range of irrigation amount is 272 to 625 mm. This is only possible if: (1) the measurement by the tipping bucket was inaccurate, or (2) the actual area of irrigation was much greater than the authors estimated. For a controlled experiment like this, the quantitative assessment of input rate is important. Please re-examine the data and revise the description.

Response to the Reviewer:

The "3-3.5 m" width reported in the previous version of the manuscript was an editing typo that was corrected accordingly in the revised manuscript. According to [1] and [2] The actual irrigated area was approximately 118 $m^2$, with length of 10-21 m and width of 3 – 5.5 m.  Furthermore, as outlined in Chapter 4.4 and Appendix D of [1], the spatial distribution variability was only measured during notably windy weather or if natural precipitation event coincided with the irrigation, while the irrigated water was seemingly evenly distributed over the plot during the calm weather. As shown in the figure below, both wind speed and wind direction were relatively stable during the entire 30-hour irrigation period, and the irrigation rate was not influenced by  the wind speed or direction. The only drastic change of wind direction and simultaneous reduction of wind speed occurred around the 19th hour of the irrigation period, almost coinciding with the break in irrigation that was made in order to conduct the bulk soil sample collection.

[Figure]

Additionally, considering that 20 m³ of water was transported to the site and spread over ~ 115 (5.5 m x 21 m) to 118 m², the total irrigation amount should ideally be in the range of 169.5 - 174 mm. As the pump that was used to move water from the 1000 L water tanks to the sprinkler system had to be submerged in water at all times, some small amount of water always had to remain at the bottom of the tank, thus slightly reducing the amount of water that was supplied to the sprinkler system. With this in mind, the measured amount of 163.6 mm seems rather reasonable and representative of the actual conditions during the experiment. 6 - 10 mm of difference (between the 169.5 – 174 range and 163.6 mm) would correspond to some 35 - 50 L  of water per tank that were lost due to the pump submergence condition.

Changes made to the manuscript text:

Lines 124 – 127: "*The sprinkler setup was installed by Määttä (2020) and maintained by Korkiakoski et al. (2022), and sprinklers were positioned so that irrigation water can be distributed evenly within the EP, covering the area of 3-5.5 m width and 10-21 m length, with a total area of approximately 118 m2 in calm weather.*"

Lines 133 – 139:" *A total amount of 163.6 mm of irrigation water was recorded by the tipping bucket precipitation gauge, which can be considered representative of the actual conditions as it is only slightly lower than the calculated amount of 169.5 – 174 mm (considering 20 m$^3$ of water and an area of 115 - 118 m$^2$). The 6 - 10 mm of total difference between the measured value and the calculated range roughly corresponds to 35 - 50 L of water  loss per tank, which is reasonable considering that water pump which moved the water from the tank to the sprinklers had to remain submerged at all times, meaning that not all water could be extracted from the tanks. Furthermore, the irrigation rate was not influenced by wind speed or direction (Fig. S7).*"

Additionally, the figure shown above, with wind speed and direction and irrigation rate during the experiment, was added to the Supplement as Figure S7.

Lines 258-260. This was Line 186 in the original manuscript. The authors simply moved the sentences to Section 3.1 without revising them. This does not address my comment on the original manuscript. Please make a meaningful revision to address the comment.

Short recap:

The original comment from the previous revision was: "*Line 186. This sentence describes the response of 35-cm depth. However, I see that the 60-cm sensor responded before the water table started to rise, but this sampler was located far above the water table. This seems contrary to the sentence. Please explain. Overall, this paragraph could use a clearer writing that is consisted with the data presented in figures.*"

The sentence in question (lines 234 – 236 of the revised manuscript) is: "*The isotope ratio of the water sampled from the pan lysimeters (blue dots and lines in Fig. 2c) in both soil profiles responded to irrigation only after the groundwater level went up to their installation depth (35 cm).*"
* * *
The new response where reviewer concern is taken account and explained more carefully:

The fact that the fastest isotopic response was observed at 60 cm depth is mentioned in the manuscript (line 218 of the revised manuscript: "*The fastest isotopic response in the EP was observed at a 60 cm depth (dashed black line in Fig. 2c) after only 5 hours.*" It is also mentioned in lines 435 – 437: "*The isotopic enrichment observed at 60 cm depth (Fig. 2b) during the irrigation, that occurred before the enrichment in upper soil layers, further shows that preferential flow pathways were active even in the early stages of the experiment.*")

The idea behind the sentence in question, regarding the pan lysimeters, was to indicate that pan lysimeters did not seem to respond at all to the "downwards" infiltration, both during the experiment (Figure 2) and during the whole observation period (Figure 4), meaning that the water originates from the

groundwater rise and/or lateral fluxes. In fact, there wasn't even any water to be collected prior to the groundwater rise. This was also previously commented on later in the manuscript (lines 437 – 439 of the previous version of revised manuscript, now in lines 458 - 460): " *On the other hand, the abrupt appearance and disappearance of large quantities of water in pan lysimeters, observed both during the irrigation experiment and the snowmelt, can be used to infer lateral flows and the near saturation of the macropore network*.") The sentence dedicated to pan lysimeter results in section 3.1 was expanded into a small section and modified to indicate the abovementioned reasoning more clearly.

Changes made to the manuscript text:

The revised paragraph (lines 243 - 252) reads as follows: "*The pan lysimeters were empty at the beginning of the experiment, indicating that no freely draining water had reached the pan lysimeter collecting bottles in the period prior to the experiment.  Furthermore, no water was found in the pan lysimeter collecting bottles during the first 20 hours of the irrigation, although approximately 100 mm of irrigation water was applied to the plot by that moment. The samples only appeared after groundwater level exceeded the installation depth of the pan lysimeters (35 cm) and the collecting bottles got "overtopped" by the rising groundwater. From that point onwards, large quantities of water were evacuated from the collecting bottles at each hour and $\delta^2$H value of pan lysimeter water rose sharply, from -52 ‰ to 2.5 ‰, reaching the most enriched values at the end of the irrigation (at 30 hours). The isotopic signal got more depleted immediately afterwards, thus displaying similar dynamics to the groundwater isotopic signal.*"

Line 384-392. This was Line 313 in the original manuscript. I am not convinced by the authors' explanation. Please present a more clear and less speculative explanation for a major shift in isotopic composition resulting from a small change in water content.

Short recap:

The original comment from the last revision was: "*Line 313. (Smaller pores in the soil matrix) is filled first. Figure 2d indicates only 5% increase in water content at 5-cm depth. This seems inconsistent with a major shift in isotopic composition depicted in Fig. 3. Does it make sense in terms of mass balance consideration? Please explain*"

The answer was: "*During the first 5 hours of the experiment, the soil moisture content in first 10 cm of depth increased by less than 10 %, while the bulk soil $d_2$H values increased by some 90 ‰ (from -77 to +11 ‰). Such $d_2$H increase could not be caused by simple mixing of the antecedent soil water with the infiltrating water, as the amount of antecedent water (~35 %) was much higher than the soil moisture increase. This indicates that the process of soil water displacement in the upper soil layers was initiated in the early stages of the experiment, as the enriched water started entering the soil matrix and altering the isotopic signal of the soil water. It should be noted that the observed soil water enrichment could not only be caused by mixing and displacement processes, as the bulk soil water also contains a certain fraction of very mobile infiltrating water, which can further skew the isotopic values towards the enriched values.*"

In the new response, the Reviewer's concern is taken account and the section was modified as follows to improve clarity. Changes to the manuscript text (lines 373 - 389):

*"During the first 5 hours of the irrigation, the average soil moisture content in top 10 cm increased by 7 % points (from 35.5 to 42.5 %), while the average bulk soil $\delta^2H$ value in top 10 cm increased by 88 ‰, from -77 to +11 ‰. During the same period, 36.2 mm of irrigation water with $\delta^2H$ value of 76.9 ‰ was applied to the plot. Assuming that 1) all additional water in the topsoil could be represented through the soil moisture content increase and 2) bulk soil $\delta^2H$ signal changes only due to simple mixing of the antecedent and newly infiltrated water; these additional 7 % points of soil moisture would have to have a $\delta^2H$ signal in the range of +460 ‰ to shift the bulk soil water $\delta^2H$ signal from -77 to +11 ‰. Alternatively, if the $\delta^2H$ value of irrigation water in the observed 5-hour period (76.9 ‰) was considered as the endmember of the newly infiltrated water, a soil moisture increase of some 41 % percentage points would be required to result in a bulk soil water enrichment of +88 ‰. Following this, it is clear that assumptions of either well-mixed conditions or piston flow are both simplifying the processes at play and that there are some limitations to using soil moisture content as a sole indicator of bulk soil $\delta^2H$ signal variability. Namely, soil moisture content cannot accurately represent soil waters of all mobilities, especially in the case of fast draining and macropore water, and furthermore cannot indicate soil water displacement process. The only way to actualize such a strong shift in the bulk soil $\delta^2H$ signal is through a combination of two processes: 1) partial soil matrix water displacement and 2) mixing with highly mobile infiltrating or macropore water. Furthermore, neither of these two processes can be clearly visible in the suction lysimeters' $\delta^2H$ values, as they do not sample the waters of very low mobility (soil matrix water contained in the smaller pores) and generally do not show an immediate response to more mobile or macropore waters.*

Line 443-460. This was Line 288 in the original manuscript. I asked the authors to present a more logical explanation about the connection between the rise of the water table and the lack of response of soil moisture at 60cm. The authors attempt to address this comment by simply removing the 60cm data from the graph. It is not acceptable to remove the essential data just because they do not fit with the interpretation. Please leave the data in the figure and come up with a revised interpretation that is consistent with the data.

Response to the Reviewer:

The authors regret the misunderstanding caused by the removal of data that shows the dynamics of soil moisture at 60 cm depth in Profile 2. The idea behind the removal of data was to be particularly strict with the inputs rather than to remove them out of convenience. The data has been now reincluded into the graphs, their effect on the main findings (soil matrix refilling/homogenization) of the study has been clearly addressed and added to the section 4.3.

Throughout the whole observation period, lasting one whole year, the soil moisture values at the Profile 2 60 cm sensor were rather high (~ 45 %) and stable, despite strong groundwater dynamics at the plot, changes observed in the other soil moisture sensor installed at 60 cm depth and isotopic changes observed in both bulk soil and soil lysimeter samples at 60 cm depth.

The observed soil moisture dynamics could have arguably been caused by sensor malfunction but could also showcase the existence of soil sections that are largely isolated from the surrounding soil matrix or

macropore network. Such decoupling can occur due to a localized soil compaction that is either a natural facet of soil heterogeneity or artificially created during the sensor installation. While the readings of this sensor should be interpreted with caution, they indicate that some patches of the soil might be decoupled from the surrounding soil at all times. However, the lack of soil moisture change does not undoubtedly prove that there are no soil water fluxes and isotopic signal changes at this location. It is entirely possible that antecedent soil water gradually gets displaced by the infiltrating water, but this effect gets obscured by the near saturation soil moisture values. The isotopic composition dynamics of soil samples collected in the vicinity of the sensor location does indicate that water mixing and displacement principally occurs at this depth. Still, the isotopic signal of samples collected at one spot is not necessarily representative of all the surrounding soil, especially considering the high spatial heterogeneity typically observed in till soils. While the full extent of potential soil decoupling cannot be quantified, it is highly unlikely that few isolated soil patches can significantly affect the conclusion of the study, i.e. a general soil isotopic homogenization that occurs as the aftermath of snowmelt.

Changes made to the manuscript text (lines 531 - 539):

*"Contrary to the observed isotopic dynamics, some hydrographic observations made both during and after the experiment, namely the soil moisture measured by 60 cm deep sensor in Profile 2, indicate that the irrigation and snowmelt events did not produce intense changes in soil moisture. The measured soil moisture value at this location was always about 45 %, regardless of hydrological conditions (see full black line in Fig. 2d and Fig 4c). While such soil moisture dynamic could have arguably been caused by sensor malfunction, it could also showcase the existence of soil sections that are largely isolated from the surrounding soil matrix or macropore network due to a localized soil compaction that is either a natural facet of soil heterogeneity or artificially created during the sensor installation. While the full extent of this potential decoupling between certain portions of soil matrix cannot be quantified, it is highly unlikely that few such isolated soil patches can significantly affect the conclusion of the study, i.e. a general soil isotopic homogenization that occurs as the aftermath of snowmelt."*

References:

[1] Tiia Määttä, 2020, METHANE FLUX CHANGES DURING IRRIGATION EXPERIMENT IN BOREAL UPLAND FOREST SOIL, Master's thesis, UNIVERSITY OF HELSINKI, FACULTY OF SCIENCE, DEPARTMENT OF GEOSCIENCES AND GEOGRAPHY, DIVISION OF GEOGRAPHY, https://helda.helsinki.fi/items/daec65cb-4f05-46ea-9474-00a1ff775f71

[2] Korkiakoski, M., Määttä, T., Peltoniemi, K., Penttilä, T., and Lohila, A.: Excess soil moisture and fresh carbon input are prerequisites for methane production in podzolic soil, Biogeosciences, 19, 2025–2041, https://doi.org/10.5194/bg-19-2025-2022, 2022.